# Derivatives Are All You Need For Learning Physical Models

## Abstract

Physics-Informed Neural Networks (PINNs) explicitly incorporate Partial Differential Equations (PDEs) into the loss function, thus learning representations that are inherently consistent with the physical system. We claim that it is possible to learn physically consistent models without explicit knowledge about the underlying equations. We propose Derivative Learning (DERL) to model a physical system by learning its partial derivatives, as they contain all the necessary information to determine the system's dynamics. Like in PINNs, we also train the learning model on the initial and boundary conditions of the system. We provide theoretical guarantees that our approach learns the true solution and is consistent with the underlying physical laws, even when using empirical derivatives. DERL outperforms PINNs and other state-of-the-art approaches in tasks ranging from simple dynamical systems to PDEs. Finally, we show that distilling the derivatives enables the transfer of physical information from one model to another. Distillation of higher-order derivatives improves physical consistency. Ultimately, learning and distilling the derivatives of physical systems turns out to be a powerful tool to learn physical models.

## 1 Introduction

Machine Learning (ML) techniques have found great success in modeling dynamical and physical systems, including Partial Differential Equations (PDEs). The growing interest around this topic is driven by the many real-world problems that would benefit from accurate prediction of dynamical systems, such as weather prediction (Pathak et al., 2022), fluid modeling (Zhang et al., 2024), quantum mechanics (Mo et al., 2022), and molecular dynamics (Behler & Parrinello, 2007). These problems require grasping the essence of the system by modeling its evolution while following the underlying physical laws. Purely data-driven models often fail at this task: while being able to approximate any function, they do not usually learn to maintain consistency with the physical dynamics of the system (Greydanus et al., 2019; Hansen et al., 2023), or fail to approximate it when only a few data points are available (Czarnecki et al., 2017). Physics-Informed Neural Networks (PINNs) (Raissi et al., 2019) emerged as an effective paradigm to learn the dynamics of a PDE, by explicitly including in the loss the PDE components evaluated using automatic differentiation (Baydin et al., 2018). This imposes physical consistency by design and allows to use PINNs in regimes where data is scarce. Unfortunately, PINNs suffer from optimization issues (Wang et al., 2022) which can lead to poor generalization (Wang et al., 2021). Other physics-inspired models exploit classical formalisms of mechanics such as Hamiltonians (Greydanus et al., 2019) and Lagrangians (Cranmer et al., 2019), but they are restricted to problems where the system is conservative and require an external solver to calculate complete trajectories.

In this paper, we propose Derivative Learning (DERL), a new approach to train neural networks using only the partial derivatives of the objective function, as they perfectly describe its evolution in time and space. Like PINNs, DERL also learns the initial and boundary conditions, as they are needed to retrieve the full solution. The idea behind learning the partial derivatives can be understood by looking at a simple dynamical system such as the pendulum, or any other system that can be described by a Cauchy problem $\dot{x}(t) = f(x(t))$, $x(0) = x_0$. The initial position and velocity give the starting point, while the time derivative of such quantities along the trajectory can completely determine the rest of the evolution. This is a consequence of the uniqueness theorem for Ordinary Differential Equations (ODEs). A similar result holds for PDEs (Evans, 2022). We

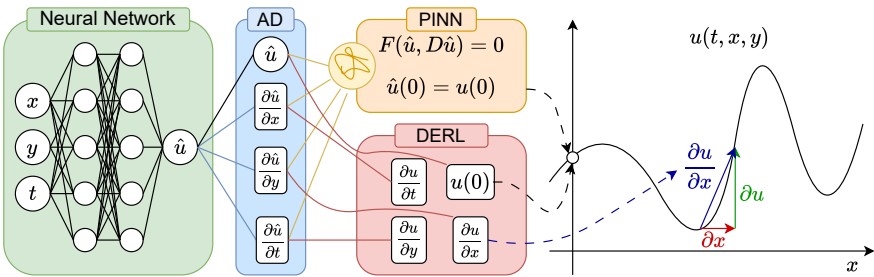

Figure 1: Comparison between how DERL and PINN learn a function $u(t, x, y)$. Partial derivatives $\frac{\partial \hat{u}}{\partial t}, \frac{\partial \hat{u}}{\partial x}, \frac{\partial \hat{u}}{\partial y}$ of the network output $\hat{u}$ are computed by Automatic Differentiation (AD) (Baydin et al., 2018). PINNs consider $\hat{u}, \nabla \hat{u}$ and entangle all derivatives when calculating the PDE residual. Instead, DERL *learns* the partial derivatives as independent targets, together with the initial or boundary condition. Derivatives can be calculated as finite differences (right-most plot).

prove that learning partial derivatives is sufficient to learn the underlying system and enables better in-domain generalization compared to data-driven approaches like supervised learning and PINNs. We also discovered that DERL enables the transfer of physical knowledge from a trained model to a student model by *distilling* the derivatives. Distilling higher-order derivatives improves the result.

The main contributions of this paper are: (a) DERL, a new methodology to learn dynamical systems and solutions to PDEs by only using the partial derivatives of the target functions, along with initial and boundary conditions (figure 1). (b) A theoretical validation of DERL which proves that the learned solution converges to the true one when the loss goes to zero. Our results hold even with empirical derivatives when the analytical ones are not available. (c) An empirical assessment of DERL on a variety of dynamical systems and PDEs, including systems of PDEs, against data-driven supervised learning, PINNs, and other state-of-the-art methodologies. (d) An application of DERL to the transfer of physical knowledge from a reference model to a student one. To the best of our knowledge, this is the first attempt at this task. We hope it can pave the way to an *incremental* and *compositional* way (Xiang et al., 2020; Soltoggio et al., 2024) of learning physical systems.

## 2 ON THE IMPORTANCE OF DERIVATIVE LEARNING

A model that learns dynamical systems must be capable of simulating the evolution of the system over a large timespan, possibly in regions or at resolutions not available during training. This makes the consistency of our model to the underlying physical laws a key factor to ensure it is a reliable and robust predictor. Concretely, given a set of evaluation points and their outputs $\{(\boldsymbol{x}_i, u(\boldsymbol{x}_i)\}, i = 1, \ldots, N$, where $\boldsymbol{x} \in \mathbb{R}^n$ and $u(\boldsymbol{x}) \in \mathbb{R}$, multiple curves pass through those points. We are interested in learning *the one curve* that is compatible with the underlying physical model.

Continuous dynamical systems are completely determined by two components: an initial state $\boldsymbol{u}_0$, and an ODE describing its evolution $d\boldsymbol{u}/dt = f(\boldsymbol{u}(t))$. The existence and uniqueness of a solution are guaranteed under hypotheses such as Lipschitz continuity of $f$. As a running example, we consider the damped pendulum. Its state is defined by the current angle and angular velocity $\boldsymbol{x}(t) = (\theta(t), \omega(t))$. The corresponding ODE reads:

$$\begin{cases} \dot{\theta} = \omega \\ \dot{\omega} = \ddot{\theta} = -\frac{g}{l}\sin(\theta) - \frac{b}{m}\omega, \end{cases} \tag{1}$$

where $g, b, m, l$ are scalar parameters that represent physical quantities such as gravity, dampening, mass, and rope length, respectively. Dynamical systems teach us an interesting fact: the initial state and the state derivatives are all that is *necessary and sufficient* to predict the full trajectory of the system. From a mathematical point of view, two functions with the same continuous derivative differ up to a constant, which is determined by some initial condition. Of greater complexity, PDEs

describe physical systems of the form:

$$
\begin{cases}
\mathcal{L}\boldsymbol{u}(t, \boldsymbol{x}) = 0 & t \in [0, T], \boldsymbol{x} \in \Omega, & \text{(PDE)} \\
\boldsymbol{u}(0, \boldsymbol{x}) = g(\boldsymbol{x}) & \boldsymbol{x} \in \Omega, & \text{(IC)} \\
\boldsymbol{u}(t, \boldsymbol{x}) = b(t, \boldsymbol{x}) & \boldsymbol{x} \in \partial\Omega, & \text{(BC)}
\end{cases}
\tag{2}
$$

where $\mathcal{L}$ is the differential operator of the ODE/PDE, which usually involves derivatives, $\Omega$ is the domain of the solution, and $\partial\Omega$ is its boundary. Well-posed problems of this kind are completely determined by the dynamics of the PDE, which determines the evolution and propagation of information in the domain, initial conditions (IC), and boundary conditions (BC).

With this in mind, it is worth asking if this concept can be leveraged for neural networks that learn an underlying physical system. If partial derivatives are sufficient to determine the evolution starting from a given state, can we train a model with just these terms to completely describe the behavior of the system? Hence, we propose to train a neural network using the following DERL loss:

$$
L(\hat{\boldsymbol{u}}, \boldsymbol{u}) = \overbrace{\lambda_u \left\| \mathrm{D}\hat{\boldsymbol{u}}(t, \boldsymbol{x}) - \mathrm{D}\boldsymbol{u}(t, \boldsymbol{x}) \right\|_{L^2([0,T]\times\Omega)}^2}^{\text{Derivative learning}} + \overbrace{\lambda_B \left\| \hat{\boldsymbol{u}}(t, \boldsymbol{x}) - b(t, \boldsymbol{x}) \right\|_{L^2([0,T]\times\partial\Omega)}^2}^{\text{Boundary cond.}} +
$$
$$
+ \underbrace{\lambda_I \left\| \hat{\boldsymbol{u}}(0, \boldsymbol{x}) - g(\boldsymbol{x}) \right\|_{L^2(\Omega)}^2}_{\text{Initial cond.}},
\tag{3}
$$

that is a combination of the $L^2$ loss on the function jacobians $\mathrm{D}\boldsymbol{u}$ or gradients $\nabla u$ when $u(\boldsymbol{x}) \in \mathbb{R}$, the BC and the IC, where $\lambda_u, \lambda_B, \lambda_I$ are hyperparameters. In practice, the $L^2$ losses are substituted by empirical ones such as the Mean Squared Error over collocation points. See Appendix A for more details. In the case of time-independent problems, such as the Allen-Cahn equation (E2) in table 1, the last term is dropped. Compared to PINNs our method is simpler to train as the network's partial derivatives have individual targets instead of being entangled together (see figure 1). Therefore, we expect better generalization capabilities. When the derivatives are not available as data, we will show that empirical derivatives obtained with finite differences on $u$ (Anderson et al., 2020) are sufficient to train the network.

**Derivative distillation for transfer of physical knowledge.** Combining knowledge from different experts is a powerful approach in deep learning (Xiang et al., 2020; Carta et al., 2024), still unexplored in the context of physical systems. We leverage DERL and distillation to transfer the physical knowledge contained in a pre-trained model to a randomly initialized student. This relieves from the burden of learning exclusively from raw data and it enables continual exchange of physical information across models. We train the student to minimize the MSE loss between the derivative $\nabla\hat{u}_T$ computed by the (frozen) teacher and the derivative computed by the (trainable) student $\nabla\hat{u}_S$ in the interior of the domain. The MSE replaces the KL divergence commonly used in distillation for classification problems. We will show that *distilling higher-order derivatives* leads to substantial improvements in the physical consistency of the final model.

## 2.1 THEORETICAL ANALYSIS

Neural networks are universal approximators in the Sobolev space $W^{m,p}(\Omega)$ of $p$-integrable functions with $p$-integrable derivatives up to order $m$ (Hornik, 1991). The only requirement is for the activation function to be continuously differentiable $m$ times (as is the $\texttt{tanh}$ function we will use). We now prove that learning the derivatives $\mathrm{D}\boldsymbol{u}$, together with the IC and BC, is sufficient to learn $\boldsymbol{u}$. We show that minimizing the loss in 3 is equivalent to minimizing the distance between our solution $\hat{\boldsymbol{u}}$ and the true one $\boldsymbol{u}$. Our first result involves ODEs and one-dimensional functions, which resembles the fundamental theorem of calculus:

**Theorem 2.1.** *Let $u(t)$ be a (continuous) function in the space $W^{1,2}([0,T])$ on the interval $[0,T]$. If $L(u, \hat{u}) \to 0$, then $u \to \hat{u}$.*

We give the proof in Appendix B.1. The generalization of this result to higher dimensions turned out to be more challenging. We state the final result as:

**Theorem 2.2.** *Let $\Omega$ be a bounded open set and $u \in W^{1,2}(\Omega)$ a function in the Sobolev space with square norm. If the neural network $\hat{u}$ is trained such that $L(\hat{u}, u) \to 0$, then $\|\hat{u} - u\|_{W^{1,2}(\Omega)} \to 0$*

*and $\|\hat{u} - u\|_{L^2(\partial\Omega)} \to 0$. To have a numerical approximation, if $L(\hat{u}, u) \leq \epsilon$, then $\|\hat{u} - u\|_{W^{1,2}(\Omega)} \leq 2(C+1)\epsilon$ for some constant $C$. Furthermore, at the limit, the two functions coincide $\hat{u}_\infty = u$ inside $\Omega$ and at the boundary.*

The proof is provided in Appendix B.2. This result can be extended to include higher-order derivatives by adding the corresponding terms to the loss $L$. We can use these theorems to prove that a neural network trained to optimize the loss in equation 3 learns the solution to a PDE. We prove the following theorem in Appendix B.3.

**Theorem 2.3.** *Let $\mathbf{u}$ be a solution to a PDE of the form of equation 2, which can be time-dependent or time-independent. A Neural Network $\hat{\mathbf{u}}$ trained to optimize $L(\hat{\mathbf{u}}, \mathbf{u})$ as in equation 3 converges to the solution of the PDE $\mathbf{u}$ and fulfills all three conditions.*

When analytical derivatives are not available, we use empirical derivatives obtained via finite differences (Anderson et al., 2020): $\frac{\partial u}{\partial x_i}(\mathbf{x}) \simeq \frac{u(\mathbf{x}+he_i)-u(\mathbf{x})}{h}$, where $e_i$ is the unit vector in the direction of $x_i$ and $h$ is a small positive real number. Appendix C shows that finite differences converge on the whole domain to the true derivatives when $h \to 0$. This guarantees that we can still approximate $u$ using empirical derivatives. We provide additional statements and remarks in Appendix B.

## 3 RELATED WORKS

**Data-driven methods.** Neural approaches are commonly used to learn physical systems and PDEs. Recurrent neural networks (Schmidt, 2019) are a prototypical example of systems that evolve through time based on their previous state. ResNets (He et al., 2016) and NODEs (Chen et al., 2019) model the residual state update as a discrete or continuous derivative, but cannot predict complete trajectories at once or require an external ODE solver. Normalizing flows (Rezende & Mohamed, 2015) model dynamical systems such as particles by learning their distribution with iterative invertible mappings. Neural operators (Li et al., 2020b) and their evolution (Li et al., 2021) act as neural networks for entire functions by mapping initial conditions or parameters to a solution via kernel operators and Fourier transforms. While these methods are powerful and find many applications to real-world problems (Pathak et al., 2022), they require a large amount of data to generalize, are computationally intensive, and do not ensure that the solution is physically consistent. Our work shares similarities with Sobolev learning (Czarnecki et al., 2017; Srinivas & Fleuret, 2018), which adds derivative learning terms to the supervised loss with application to reinforcement learning and machine vision tasks. Instead, we are the first to consider a pure derivative approach and to apply it to physical and dynamical systems.

**Physics-inspired and Physics-Informed methods.** PINNs (Raissi et al., 2019) incorporate PDEs that describe the underlying physical system directly into the loss (as well as the $L^2$ residual), using automatic differentiation (Baydin et al., 2018). PINNs reduce the amount of training data needed and increase the physical consistency of the solution. However, PINNs suffer from optimization problems (Wang et al., 2021) and can fail to reach a minimum of the loss (Sun et al., 2020). To alleviate these issues, there exist methodologies to simplify the objective (Sun et al., 2020) or to choose the collocation points where the PDE residual is evaluated (Zhao, 2021; Lau et al., 2024). Both approaches are problem-specific and add complexity to the optimization process. Models inspired by physics exploit the formalisms of Hamiltonian (Greydanus et al., 2019) and Lagrangian (Cranmer et al., 2019) mechanics to be inherently consistent with the properties of the system, but they require the system to be conservative and to be formulated explicitly or implicitly in the Lagrangian or Hamiltonian formalism, which is generally not the case.

**Exact consistency by design.** Adopting ad-hoc architectural choices for the model allows the learning process to be exactly consistent with the physical properties of the underlying system. Hansen et al. (2023) takes a probabilistic approach by modeling the predicted distribution of the solution and by projecting it on a subspace that is conservative. However, they only consider global conservation of quantities, and the approach is only tested on one-dimensional systems. Neural Conservation Laws (NCL) (Richter-Powell et al., 2022) take advantage of the mathematical theory of geometric analysis to make the output of the model divergence-free by design. However, NCL is very slow and it is only built for systems that are described by a divergence-free equation. Similarly,

Table 1: Summary of the tasks we consider in the experiments.

| Experiment | Equation | Description |
|:---:|:---:|:---:|
| **Pendulum** | (E1)  $\dot{\theta} = \omega, \quad \dot{\omega} = -\frac{g}{l}\sin(\theta) - \frac{b}{m}\omega$ | ODE |
| **Allen-Cahn** | (E2)  $\lambda(u_{xx} + u_{yy}) + u(u^2 - 1) = f$ | Second-order PDE |
| **Continuity** | (E3)  $\frac{\partial \rho}{\partial t} + \nabla \cdot (\boldsymbol{v}\rho) = 0$ | Time-dependent PDE |
| **Navier-Stokes** | (E4.M)  $\dfrac{\partial \boldsymbol{u}}{\partial t} - \mu\Delta\boldsymbol{u} + \rho[\mathrm{D}\boldsymbol{u}]\boldsymbol{u} = -\nabla p,$ 
 (E4.I)  $\nabla \cdot \boldsymbol{u} = 0$ | System of PDEs |
| **KdV distillation** | (E5)  $u + uu_t + \nu u_{xxx} = 0$ | Third-order PDE |
| **NCL distillation** | (E6.C)  $\dfrac{\partial \rho}{\partial t} + \nabla \cdot (\rho\boldsymbol{u}) = 0, \quad$ (E6.I)  $\nabla \cdot \boldsymbol{u} = 0$ 
 (E6.M)  $\dfrac{\partial \boldsymbol{u}}{\partial t} + [\mathrm{D}\boldsymbol{u}]\boldsymbol{u} + \dfrac{\nabla p}{\rho} = 0$ | System of PDEs |

Torres et al. (2024) propose a divergence-free normalizing flow, but that requires invertible mappings and it can only work with densities. Again, this approach works only with divergence-free fields and cannot be extended to include other equations in the framework.

## 4  EXPERIMENTS

We validate our approach on a set of dynamical systems and physical PDEs. Table 1 summarizes the tasks we consider. For each experiment, we compare 4 training methods on a Multi-Layer Perceptron (MLP, Goodfellow et al. (2016)): our **DERL** that learns $\mathrm{D}\boldsymbol{u}$, **Output Learning (OUTL)** that learns $u$, **PINN** (Raissi et al., 2019), and **Sobolev learning (SOB)** (Czarnecki et al., 2017). Details on these models are in Appendix A. The approaches differ in the way they learn the solution in the interior of the domain, while they enforce the same IC and BC.

To evaluate the accuracy of the prediction, we compute the $L^2$ distance between the true function $\boldsymbol{u}$ and the estimate $\hat{\boldsymbol{u}}$. To measure the physical consistency of a model, we either compute the $L^2$ norm of the PDE residual of the network as for PINNs (Raissi et al., 2019) or the $L^2$ distance between the true and the learned field in the phase space $\dot{\boldsymbol{x}}$ for dynamical systems (Strogatz, 2019).

For each experiment and each methodology, we tuned the hyperparameters for the respective losses and the learning rate independently, to obtain the top performance for each model. We report the implementation details in Appendix A and describe the tuning process in Appendix D.1. Specific experiment remarks along with many additional results are available in Appendix E.

### 4.1  DAMPED PENDULUM

We consider the dynamical system of a damped pendulum with state equation 1, which we briefly considered in Section 2. Additional details on the experimental setup are available in Section E.1. We sampled a total of 50 trajectories from different starting conditions $(\theta_0, \omega_0)$: 30 reserved for training, 10 for validation, and 10 reserved for testing. Each trajectory was sampled every $\Delta t = 0.01s$. For each method, we used a 4-layer MLP with 20 units per layer and $\tanh$ activation, trained for 200 epochs with batch size 32. The model takes as input the time and the initial condition $(t, \theta_0, \omega_0)$ and is trained to predict the corresponding position and angular speed $(\theta(t; \theta_0, \omega_0), \omega(t; \theta_0, \omega_0))$ at time $t$. Initial states are sampled randomly in $[-\frac{\pi}{2}, \frac{\pi}{2}] \times [-1.5, 1.5]$.

The task consists in learning the vector field in the phase space, that is to learn the derivatives $(\dot{\theta}, \dot{\omega})$ that best match the true ones from equation 1. The loss on the PDE residual measures the distance between $\dot{\theta}$ and $\omega$, while the field error measures the distance between the derivative of *each* network's output and the true derivative.

Figure 2a shows a direct comparison of the errors in the learned field $(\dot{\theta}, \dot{\omega})$ in the domain. We computed the local error on grid points for each method. Then, for each method, we computed the

Table 2: Results for the damped pendulum experiment. We report the loss on the state and its derivatives on testing trajectories, the $L^2$ distance between true and predicted fields, the PDE residuals, and the PDE residual for the initial condition differentiability. Fields and PDE residuals are calculated at $t = 0$. Bold denotes the best model.

| Model | State loss | Derivative loss | Field error | PDE res. | Init PDE res. |
|---|---|---|---|---|---|
| **DERL** (ours) | 0.025719 | **0.011121** | **0.28006** | **0.23656** | **0.58299** |
| **OUTL** | 0.017737 | 0.032557 | 0.98303 | 0.76095 | 1.7741 |
| **PINN** | 0.018198 | 0.018843 | 0.62468 | 0.47643 | 1.3161 |
| **SOB** | **0.015823** | 0.014177 | 0.42762 | 0.34912 | 0.91480 |

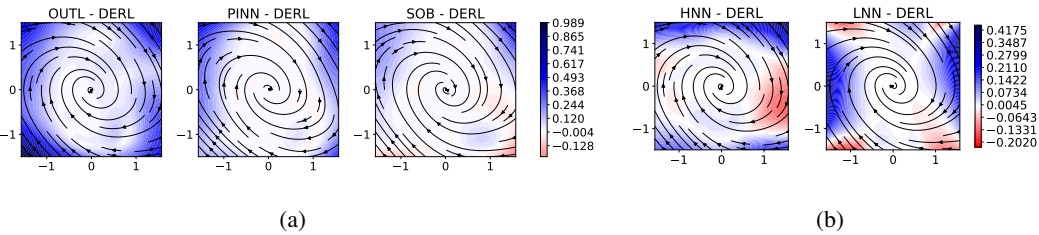

(a)                                                                    (b)

Figure 2: Pendulum experiment. $L^2$ error difference in the learned field at $t = 0$ between each methodology and DERL. The blue area is where DERL performs better than the comparison.

difference between its errors and DERL's error, such that positive values (blue) are where DERL performs better. Table 2 shows the errors on the test trajectories, the $L^2$ distance between the true and predicted fields, and the ODE residual $L^2$ norm. These results show the generalization ability of the approaches for *unseen* starting conditions. The initial condition PDE (see Appendix E.1.2 for details) measures the regularity of the model with respect to $(\theta_0, \omega_0)$ and, in particular, if the trajectories are differentiable up to the first order on these variables. DERL significantly outperforms the other approaches. DERL is therefore the best at generalizing equation E1 to new initial conditions, while OUTL and PINN are the worst-performing ones.

For this task, we adapted Lagrangian Neural Networks (LNN) (Cranmer et al., 2019) and Hamiltonian Neural Networks (HNN) (Greydanus et al., 2019) to the damped pendulum case. We train them on the conservative part of the field (see E.1.5 for details), where they excel as they are specifically designed for conservative fields. Unlike DERL, they also require trajectories to be calculated through external solvers. Remarkably, DERL outperforms both LNN and HNN (figure 2b): DERL scores a field error of $0.28006$, against $0.44699$ and $0.44277$ for LNN and HNN, respectively.

## 4.2 ALLEN-CAHN EQUATION

We now move to PDEs. The Allen-Cahn (equation (E2) in table 1) is a time-independent non-linear PDE, with $\lambda = 0.01$ and analytical solution $u_{\text{true}} = \sin(\pi x)\sin(\pi y)$. The BC and the external force $f$ are calculated from $u_{\text{true}}$ using the PDE. We sample the solution on a grid with $\Delta x = \Delta y = 0.02$ both inside the

Table 3: Results for the Allen-Cahn equation: $L^2$ distances from the ground truth. The best results are in bold.

| Model | $\|\hat{u} - u_{\text{true}}\|_2$ | $\|\nabla\hat{u} - \nabla u_{\text{true}}\|_2$ | PDE $L^2$ **norm** |
|---|---|---|---|
| **DERL** (ours) | **0.010380** | **0.033228** | **0.0096173** |
| **OUTL** | 0.018174 | 0.15132 | 0.030412 |
| **PINN** | 030950 | 0.11028 | 0.028795 |
| **SOB** | 0.015356 | 0.048076 | 0.016461 |

domain and at the boundary. Partial derivatives are calculated analytically from the true solution. The MLP has $4$ layers with $50$ units and `tanh` activation and is trained for $100$ epochs with batch size $32$. Model selection details and results are provided in Appendix E.2.

Table 3 shows the $L^2$ distance between the true and predicted solution (including partial derivatives), as well as the $L^2$ PDE residual, which is equivalent to the error in the learned forcing $f$. As for the

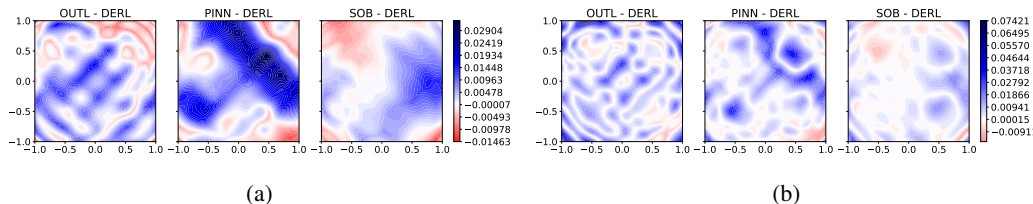

(a)                          (b)

Figure 3: Allen-Cahn experiment: (a) $u$ error comparison between DERL and the other methodologies. (b) PDE residual comparison between DERL and the other methodologies. Blue regions are where DERL performs better than the comparison.

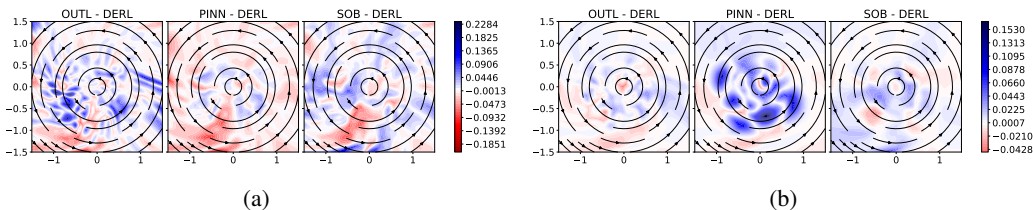

(a)                          (b)

Figure 4: Continuity equation experiment: (a) Comparison of $L^2$ PDE residual on the domain at $t = 5$. Differences between the other methods' residuals and DERL. Blue regions are where we perform better (b) Same plot but with $L^2$ distance w.r.t. the true solution $\rho_{\text{true}}$.

damped pendulum, figure 3a shows the error difference on the solution $u$ between DERL and the other approaches. Similarly, figure 3b shows the error difference for the PDE residual. Calculations and color definitions for the plots are the same as in the pendulum experiment. DERL outperforms all the other approaches in learning the true solution $u_{\text{true}}$. As expected, DERL also approximates best the derivatives. Finally, DERL satisfies the PDE 3 times better than OUTL and PINN. The fact that the second best method is SOB shows again the effectiveness of learning the derivatives. This is also an empirical confirmation of our theoretical results. In the case where data about $u$ in $\Omega$ is not available, the models have to propagate information from the boundary to the interior of the domain. PINN underperforms in this setup, while DERL successfully solves the task, while also showing a stronger consistency with the second-order derivatives PDE.

### 4.3 CONTINUITY EQUATION

As a first example of time-dependent PDE, we consider the continuity equation (equation (E3) in table 1), used to model conservation laws. The domain of the unknown density $\rho(t, x, y)$ is the 2D plane region $[-1.5, 1.5]^2$ for $t \in [0, 10]$. The velocity field $\boldsymbol{v}$ is the rotational divergence-free field $\boldsymbol{v}(x, y) = (-y, x)$. The density is null at the boundary and the IC is given by 4 Gaussian densities (see Appendix E.3), as in Torres et al. (2024). The reference solution is calculated using the finite volumes method (Ferziger & Peric, 2001) on a regular grid characterized by $\Delta x = \Delta y = 0.01$ and time discretization of $\Delta t = 0.001$.

The MLP has the same architecture as the Allen-Cahn experiment and is trained for 200 epochs with batch size 128. Partial derivatives were calculated by finite difference approximation with $\Delta x = \Delta y = \Delta t = 0.01$, without any interpolation. The MLP takes as input $(t, x, y)$ and predicts $\rho(t, x, y)$. See E.3 for further details and results, where we also report results for the interpolated data case.

Table 4 reports the $L^2$ error on the solution $\rho(x, y, t)$ and the PDE residual $L^2$ norm on the whole time-space domain. Figure 4a shows the comparison between DERL and the other methods

Table 4: Continuity equation results. $L^2$ norms w.r.t. the ground truth. Best models are in bold.

| Model | $\|\rho - \rho_{\text{true}}\|_2$ | PDE $L^2$ norm |
|---|---|---|
| **DERL** (ours) | 0.028827 | 0.073379 |
| **OUTL** | **0.027932** | 0.12411 |
| PINN | 0.088850 | **0.041071** |
| SOB | 0.052511 | 0.092739 |

on the PDE residual norm and the distance from the true solution at $t = 5$. We found DERL to be the most effective method: the distance from the ground truth is comparable to OUTL and two orders of magnitude smaller than all the others. Plus, DERL is also second best for PDE consistency, closely following PINNs (which, we recall, impose the PDE as a hard constraint). However, PINNs completely fail to propagate the solution through time correctly, as reported by both table 4 and figure 4b at $t = 5$. These results for PINNs are aligned with Wang et al. (2022), where the authors conjectured that PINN's gradients are biased towards high values of $t$ with the model failing to propagate information from the initial solution. DERL, on the other hand, correctly learns the complete solution *without ever seeing any data for* $t > 0$. The continuity equation is a strong example showing how derivatives are all that is needed to learn the system. It is also interesting to see how SOB performed worse in all metrics. This is probably due to the more challenging and conflicting loss terms, which require Sobolev to learn both the derivatives and the outputs at the same time.

## 4.4 NAVIER STOKES EQUATIONS

We now consider *a system of time-dependent PDEs with multiple outputs*, the most challenging among our tasks. As in Raissi et al. (2019), we consider the transient 2D Navier-Stokes equations, made of the momentum equation (E4.M) and the incompressibility equation (E4.I) (table 1), with uniform density $\rho = 1$ and viscosity $\mu = 10^{-3}$. The unknowns are the 2D fluid's velocity $\boldsymbol{u}$ and the pressure $p$. For the setup, please refer to Appendix E.4.1. Following Raissi et al. (2019), we considered the region adjacent to the right side of the circular obstacle. The domain is $(x, y) \in [0, 1.7] \times [0, 0.41]$ for times $t \in [0, 2]$. IC, BC and internal data are given by the true solution obtained with the finite volumes method (Anderson et al., 2020). The training data has a grid size of $\Delta x = \Delta y = \Delta t = 0.01$. The MLP is made of 8 tanh layers with 128 units. The training lasted for 200 epochs with a batch size of 512. In Raissi et al. (2019), the solution is parametrized such that equation (E4.I) is satisfied by design. Here, the network takes as input $(t, x, y)$ and predicts $\boldsymbol{u}$ and $p$ so that both equations have to be learned at the same time.

Table 5: Results for the Navier Stokes experiments. $L^2$ error on the final solution, $L^2$ norms of the residuals of the 2 PDEs. Norms are calculated across the entire time-space domain. Best model in bold.

| Model | $L^2$ error | (E4.M) $L^2$ norm | (E4.I) $L^2$ norm |
|---|---|---|---|
| **DERL** (ours) | 0.021687 | **0.36237** | 0.30337 |
| **OUTL** | **0.011950** | 0.81446 | 0.33483 |
| **PINN** | 0.63828 | 6.9591 | 3.9806 |
| **SOB** | 0.015714 | 0.60096 | **0.29979** |

Numerical results are in table 5. The PINN learns the IC at $t = 0$ but fails to propagate it to later time steps (see Appendix E.4), thus learning a solution which diverges from the true one by a large margin. DERL outperforms all other approaches in the momentum equation residual (E4.M), the most challenging equation. All models except PINN perform similarly on (E4.I). We again stress how DERL achieves these results without having access to the solution in the interior of the time-space domain. We provide results with randomly sampled points and empirical derivatives in Appendix E.4.3.

## 4.5 TRANSFERRING PHYSICAL INFORMATION ACROSS MODELS

We investigate the ability of DERL to transfer physical constraints from a pre-trained model to a student model by *distilling* the teacher's derivatives. To the best of our knowledge, we are the first to leverage distillation for this purpose. We experimented with a PINN on the Korteweg-de Vries (KdV) equation and a Neural Conservation Laws (NCL) model (Richter-Powell et al., 2022) on the Euler equations.

### 4.5.1 PINN DISTILLATION

We consider the Korteweg-de Vries (KdV) equation, a third-order non-linear PDE (equation (E5) in table 1), with $\nu = 0.0025$. The IC is $u(0, x) = \cos(\pi x)$ with periodic BC for $u$ and $u_x$. A PINN is trained on a reference solution on a grid with $\Delta x = \Delta t = 0.005$. The resulting model is treated as the teacher model. The architecture of the student is the same as the teacher. More details are provided in Appendix E.5.1. Since this is a third-order PDE, we are also interested in understanding

Table 6: Results for the KdV equation, PINN distillation. Results are empirical $L^2$ norms over the time-space domain. Derivative and Hessian losses are computed with respect to the PINN. Best model(s) in bold, second best underlined.

| Model | $\|\hat{\mathbf{u}} - \mathbf{u}_{\text{true}}\|_2$ | $\|\nabla\hat{\mathbf{u}} - \nabla\mathbf{u}_{\text{PINN}}\|_2$ | $\|\mathbf{H}_{\hat{\mathbf{u}}} - \mathbf{H}_{\text{PINN}}\|_2$ | PDE loss | BC loss |
|---|---|---|---|---|---|
| PINN (teacher) | 0.037171 | / | / | 0.16638 | 0.33532 |
| **DERL** (ours) | 0.038331 | 0.098188 | 3.9872 | 0.32480 | 0.014197 |
| **HESL** (ours) | **0.037380** | 0.065454 | 1.1662 | **0.19153** | 0.014220 |
| **DER+HESL** (ours) | 0.038524 | **0.041988** | **0.85280** | 0.19317 | 0.031850 |
| **OUTL** | 0.038589 | 0.22580 | 31.582 | 17.366 | **0.012830** |
| **SOB** | 0.037447 | 0.10097 | 4.0967 | 0.38523 | 0.013644 |
| **SOB+HES** | 0.041353 | 0.13119 | 3.2684 | 0.23184 | 0.016222 |

the impact of higher-order derivatives on the performance, both in terms of physical consistency (the PDE residual norm) and in terms of matching the true solution $u_{\text{true}}$ or the BC. We implemented **Hessian learning (HESL)**, which learns the Hessian matrices of the teacher model. We also explored its combination with DERL (DER+HESL, which learns both $\nabla u_{\text{PINN}}$ and $\boldsymbol{H}_{\text{PINN}}(x, t)$) and Sobolev (SOB+HES). In this last case, we use the approximation of the Hessian matrix as suggested in Czarnecki et al. (2017), while for HESL we use the full Hessian of the network, at the cost of an increase in computational time (see Section D.2 for further details). We remark that each methodology has been tuned individually to find the hyperparameters that provide the best approximation of the true solution $u_{\text{true}}$. Table 6 reports the $L^2$ distance to the true solution, relevant distillation metrics, BC errors, and the PDE residual norm on the domain. First, we notice that all methods successfully learned to approximate the solution $u_{\text{true}}$ with the same performance as the teacher model, with SOB+HES being the worst by a small margin. We observe that (a) distilling a PINN can lead to noticeable **performance improvements**. In particular, our HESL and DER+HESL achieved the same PDE residual norm and distance to the true solution of the teacher model but with a BC loss smaller by at least one order of magnitude. This may be due to the easier optimization objective of the student models. (b) OUTL fails to distill the physical knowledge of an architecturally identical PINN, as the PDE residual is two orders of magnitude larger than any other model. Interestingly, the best methods in terms of physical consistencies are those that did not see the values of $u$ directly. (c) When boundary conditions are available for both $u, u_x$, **second order derivatives are sufficient** to learn the true solution with PDE consistency comparable to PINN. (d) HESL and DER+HESL showed the best overall results. This tells us that adding one more derivative helps learning and distilling high-order PDEs. Furthermore, approximating Hessians as in SOB+HES (Czarnecki et al., 2017) slightly reduces the performance compared to using the full Hessian of the network.

### 4.5.2 NCL DISTILLATION

Table 7: Results for the NCL distillation experiment. $L^2$ distance between the NCL and our distilled solutions, $L^2$ norms of the residuals of the 2 PDEs. Norms are calculated across the entire time-space domain. The best results are in bold.

| Model | $L^2$ error | (E6.M) $L^2$ norm | (E6.I) $L^2$ norm |
|---|---|---|---|
| **DERL** (ours) | 0.015287 | **0.28566** | **0.095044** |
| **OUTL** | 0.022247 | 0.52101 | 0.17159 |
| **SOB** | **0.013282** | 0.28620 | 0.10134 |

We perform knowledge distillation with the NCL architecture (Richter-Powell et al., 2022), considering the Euler equations for incompressible inviscid fluids with variable density in the 3D unit ball (3 PDEs). The system is made of the mass conservation (6.D), incompressibility (6.I) and momentum (6.M) equations (table 1), where $\rho$ is the density, $\boldsymbol{u}$ the velocity and $p$ the pressure. IC and BC are from Richter-Powell et al. (2022). Equation (E6.D) is guaranteed by the specific architecture of the NCL model, while (E6.M) and (E6.I) are learned with the usual PINN style PDE residual loss. The teacher and student models' setup is the same as in Richter-Powell et al. (2022).

As Equation (E6.D) is guaranteed by NCL design, table 7 reports results for Equations (E6.M) and (E6.I). Although minimal, distillation reported an improvement in the BC (from 0.09 for the

teacher to around 0.07 for the students). DERL and SOB outperform OUTL, with almost 50% error reduction in each metric. Even with a different architecture like NCL, these results highlight the effectiveness of derivative distillation for the transfer of physical knowledge across models.

## 5    CONCLUSION

We proposed DERL, a methodology to learn dynamical and physical systems using only the partial derivatives of the solution together with the initial and boundary conditions of the problem. We showed theoretically and experimentally that our method successfully learns the solution to a problem and remains consistent with its physical constraints, outperforming PINN and other supervised learning methods. We also found DERL to be effective in transferring physical knowledge across models through distillation, and we showed how higher-order derivatives can contribute to an increased physical consistency of the learned solution. Much like other deep learning applications (e.g. natural language processing, computer vision), in the future learning physical and dynamical systems may not require always starting from scratch. Instead, the learning process may be based on the continual composition and integration of physical information across different models. A more general and flexible paradigm that is still underexplored to date, for which DERL can be a good candidate to provide the foundational mechanisms and core principles.

### ETHICS STATEMENT

We do not identify any ethical concerns or societal risks for this work. The datasets do not contain any sensitive or privacy-related information. Our work does not involve human subjects or crowdsourcing methods.

### REPRODUCIBILITY STATEMENT

The code to reproduce all experiments is available as supplementary material. The main text and the appendix (A and D.) provide all the details about the experiments setup.

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

## A    IMPLEMENTATION DETAILS

In this Section, we give additional details on the implementation of the models and their respective losses. We start by defining the individual loss components in the most general case. Each experiment will feature its required terms based on the problem definition.

**Sobolev norms.** Let $u(\boldsymbol{x}) : \Omega \to \mathbb{R}$ be the function we want to learn. In the case of $u$ taking values in $\mathbb{R}^D$, we can take the sum over the vector components. The main loss terms are those linked to learning the solution in the domain $\Omega$. The distance in the Sobolev space $W^{m,2}(\Omega)$ between the function $u$ and our Neural Network $\hat{u}$ is defined as

$$\|u - \hat{u}\|_{W^{m,2}(\Omega)}^2 = \|u - \hat{u}\|_{L^2(\Omega)}^2 + \sum_{l=1}^{m} \left\| \mathrm{D}^l u - \mathrm{D}^l \hat{u} \right\|_{L^2(\Omega)}^2, \tag{4}$$

where $\mathrm{D}^l u$ is the $l$-th order differential of $u$, i.e. the gradient or Jacobian for $l = 1$ and the Hessian for $l = 2$. Practically, these squared norms are approximated via Mean Squared Error (MSE) on a dataset $\mathcal{D}_d$ of collocation points $\boldsymbol{x}_i \in \Omega, i = 1, \ldots, N_d$ with their respective evaluations of the function and its differentials:

$$\mathcal{D}_d = \left\{ (\boldsymbol{x}_i, u(\boldsymbol{x}_i), \mathrm{D}u(\boldsymbol{x}_i), \ldots, \mathrm{D}^l u(\boldsymbol{x}_i)) \right\}, \quad \boldsymbol{x}_i \in \Omega, \quad i = 1, \ldots, N_d. \tag{5}$$

The MSE is then calculated as

$$\|u - \hat{u}\|_{L^2(\Omega)}^2 \simeq \frac{1}{N_d} \sum_{i=1}^{N_d} \left( u(\boldsymbol{x}_i) - \hat{u}(\boldsymbol{x}_i) \right)^2$$

$$\left\| \mathrm{D}^l u - \mathrm{D}^l \hat{u} \right\|_{L^2(\Omega)}^2 \simeq \frac{1}{N_d} \sum_{i=1}^{N_d} \left( \mathrm{D}^l u(\boldsymbol{x}_i) - \mathrm{D}^l \hat{u}(\boldsymbol{x}_i) \right)^2, \quad l = 1, \ldots, m \tag{6}$$

Each of the models will feature one or more terms in equation 6. In particular, OUTL uses only the first, our methodologies use only the second with one or more values of $l$, and SOB uses both. In the case of time-dependent problems, it is sufficient to consider the augmented domain $\tilde{\Omega} = [0, T] \times \Omega$ and collocation points that are time-domain couples $(t_i, \boldsymbol{x}_i), i = 1, \ldots, N_d$. The models' derivatives are calculated using Automatic Differentiation (Baydin et al., 2018) using the python PyTorch package (Paszke et al., 2019). In the case of SOB, the derivative terms of order $l \geq 2$ are approximated using discrete difference expectation over random vectors (see Czarnecki et al. (2017) and Rifai et al. (2011) for more details).

**PDE residuals.** For PINN learning and the evaluation of the physical consistency of a model, we adopt the $L^2$ norm of the corresponding PDE residual, as in Raissi et al. (2019). Let the problem be defined by a Partial Differential Equation $\mathcal{L}u(\boldsymbol{x}) = f(\boldsymbol{x})$, where $\mathcal{L}$ is a differential operator

$$\mathcal{L}u(\boldsymbol{x}) = a_0(\boldsymbol{x})u(\boldsymbol{x}) + \sum_{i=1}^{n} a_i(\boldsymbol{x}) \frac{\partial u}{\partial x_i}(\boldsymbol{x}) + \sum_{i=1}^{n} \sum_{j=1}^{n} a_{ij}(\boldsymbol{x}) \frac{\partial^2 u}{\partial x_i \, \partial x_j}(\boldsymbol{x}) + \ldots \tag{7}$$

The PDE residual measure we consider is the $L^2$ norm of $\mathcal{L}\hat{u} - f$, which is approximated again as the MSE over collocation points dataset $\mathcal{D}_d$ defined above. In this case, there is no supervised target and the operator is applied to the model using AD, obtaining the loss term

$$\|\mathcal{L}\hat{u} - f\|_{L^2(\Omega)}^2 = \frac{1}{N_d} \sum_{i=1}^{N_d} \left( \mathcal{L}\hat{u}(\boldsymbol{x}_i) - f(\boldsymbol{x}_i) \right)^2. \tag{8}$$

This loss will be used to measure the consistency of our model to the physics of the problem or the PDE in general in a strong sense since it contains the derivatives of the network itself. For PINN learning, this is the term that substitutes equation 6 to learn the solution in the domain.

**Boundary (and Initial) conditions.** To learn the function $u(\boldsymbol{x})$, as per theorem 2.2, we need to provide some information at the boundary $\partial\Omega$ as well. Otherwise, we can only conclude that $u(\boldsymbol{x})$ and $\hat{u}(\boldsymbol{x})$ differ by a possibly non-zero constant. In one dimension, this reminds us of the indefinite integral $F(x) = \int f(x) + C$, which is defined up to a constant $C$ determined by some condition on the function integral $F$. To measure the $L^2$ distance between the true function and our approximation on the boundary, we employ the usual $L^2(\partial\Omega)$ norm approximated via MSE loss on a dataset of collocation points

$$\mathcal{D}_b = \left\{ (\boldsymbol{x}_i, u(\boldsymbol{x}_i) = b(\boldsymbol{x}_i)) \right\}, \quad \boldsymbol{x}_i \in \partial\Omega, \quad i = 1, \ldots, N_b. \tag{9}$$

The norm is then approximated as

$$\|u - \hat{u}\|_{L^2(\partial\Omega)} \simeq \frac{1}{N_d} \sum_{i=1}^{N_b} \left( b(\boldsymbol{x}_i) - \hat{u}(\boldsymbol{x}_i) \right)^2. \tag{10}$$

In the case of time-dependent problems, we consider the augmented domain $\tilde{\Omega} = [0, T] \times \Omega$. In this case, the collocation points of the boundary conditions are time-space couples $(t_i, \boldsymbol{x}_i) \in [0, T] \times \partial\Omega$. In this case, initial conditions $u(0, \boldsymbol{x}) = g(\boldsymbol{x})$ are also required as part of the extended boundary and are learned with

$$\|u(0, \boldsymbol{x}) - \hat{u}(0, \boldsymbol{x})\|_{L^2(\Omega)} \simeq \frac{1}{N_i} \sum_{i=1}^{N_i} \left( g(\boldsymbol{x}_i) - \hat{u}(0, \boldsymbol{x}_i) \right)^2. \tag{11}$$

In theory, the augmented boundary should contain the values of $u(T, \boldsymbol{x})$ at the final time $T$. As these are usually not present in PDE problems or classical PINN training procedures (Raissi et al., 2019), we decided not to include them. As shown in our experiments in Section 4 and Appendix E, this led to interesting results and differences between the models. PINN learning failed to propagate the solution from $t = 0$ to higher times, while the other models did not suffer from this issue.

Each of the above components, when present, will have its weight in the total loss, which is one hyperparameter to be tuned. Below we report details on the baseline models and their losses.

**Details for the baseline models.** The baselines used in the main text comprehend:

- OUTL, that is supervised learning of the solution $u$.
- PINN (Raissi et al., 2019), which is based on the optimization of the PDE residuals of the network in equation 8 with automatic differentiation (Baydin et al., 2018).
- SOB, that is Sobolev learning, first introduced in Czarnecki et al. (2017), which considers the both the losses on $u$ and its derivatives, as in equation 6.

All of these learn IC and BC as well.

**Loss comparison.** We report here the losses used for each methodology to have a clean comparison between them. For space reasons, we write the true norms instead of the MSE approximations described above. Each norm is in $L^2(\Omega)$ or $L^2([0, T] \times \Omega)$ based on the setting.

$$
\begin{aligned}
\text{OUTL: } & L(u, \hat{u}) = \lambda_D \|u - \hat{u}\|^2 + \lambda_I \text{IC} + \lambda_B \text{BC} \\
\text{DERL: } & L(u, \hat{u}) = \lambda_D \|\text{D}u - \text{D}\hat{u}\|^2 + \lambda_I \text{IC} + \lambda_B \text{BC} \\
\text{DER+HESL: } & L(u, \hat{u}) = \lambda_D \left( \|\text{D}u - \text{D}\hat{u}\|^2 + \|\text{D}^2 u - \text{D}^2 \hat{u}\|^2 \right) + \lambda_I \text{IC} + \lambda_B \text{BC} \\
\text{HESL: } & L(u, \hat{u}) = \lambda_D \|\text{D}^2 u - \text{D}^2 \hat{u}\|^2 + \lambda_I \text{IC} + \lambda_B \text{BC} \\
\text{SOB: } & L(u, \hat{u}) = \lambda_D \left( \|u - \hat{u}\|^2 + |\text{D}u - \text{D}\hat{u}|^2 \right) + \lambda_I \text{IC} + \lambda_B \text{BC} \\
\text{SOB+HES: } & L(u, \hat{u}) = \lambda_D \left( \|u - \hat{u}\|^2 + \|\text{D}u - \text{D}\hat{u}\|^2 + \|\text{D}^2 u - \text{D}^2 \hat{u}\|^2 \right) + \lambda_I \text{IC} + \lambda_B \text{BC} \\
\text{PINN: } & L(u, \hat{u}) = \lambda_D \|\mathcal{L}u - f\|^2 + \lambda_I \text{IC} + \lambda_B \text{BC}
\end{aligned}
\tag{12}
$$

where BC and IC are respectively given by equation 10 and equation 11.

# B PROOFS OF THE THEORETICAL STATEMENTS

## B.1 PROOF OF THEOREM 2.1

*Proof.* Since $u(t) \in W^{1,2}([0, T])$, $u$ is actually Holder continuous or, to be precise, it has a Holder continuous representative in the space (Maz'ya, 2011). Since $L(u, \hat{u}) \to 0$, we have that $\hat{u}'(t) \xrightarrow{L^2([0,T])} u'(t)$ and $\hat{u}(0) \to u(0)$ (the initial conditions are just one point). If $v = u - \hat{u}$, we have that $v' \xrightarrow{L^2} 0$ and $v$ converges to a function a.e. constant, which we can suppose to be continuous as above. Then, since $v(0) = 0$, we have that $v \equiv 0$ and $\hat{u} \equiv u$. $\qquad\square$

## B.2 PROOF OF THEOREM 2.2

We begin by stating Poincaré inequality (see also Evans (2022) and Maz'ya (2011)).

**Theorem B.1** (Evans (2022) Section 5.6, theorem 3 and Section 5.8.1, theorem 1). *Let $1 \leq p < \infty$ and $\Omega$ be a subset bounded in at least one direction. Then, there exists a constant $C$, depending only on $\Omega$ and $p$, such that for every function $u$ of the Sobolev space $W_0^{1,p}(\Omega)$ of functions null at the boundary, it holds:*

$$\|u\|_{L^p(\Omega)} \leq C\|\nabla u\|_{L^p(\Omega)} \tag{13}$$

*In case the function is not necessarily null at the boundary $u \in W^{1,p}(\Omega)$ and $\Omega$ is bounded, we have that*

$$\|u - (u)_\Omega\|_{L^p(\Omega)} \leq C\|\nabla u\|_{L^p(\Omega)} \tag{14}$$

*where $(u)_\Omega = \frac{1}{|\Omega|} \int_\Omega u \, \mathrm{d}x$ is the average of $u$.*

To prove one of our results we will also need the following generalization of Poincaré's inequality.

**Theorem B.2** (Maz'ya (2011) Section 6.11.1, corollary 2). *Let $\Omega \subseteq \mathbb{R}^n$ be an open set with finite volume, $u \in W^{1,p}(\Omega)$ such that the trace of $u$ on the boundary $\partial\Omega$ is $r$-integrable, that is $\|u\|_{L^r(\partial\Omega)} < \infty$. Then, for every $r, p, q$ such that $(n - p)r \leq p(n - 1)$ and $q = \frac{rn}{n-1}$, it holds*

$$\|u\|_{L^q(\Omega)} \leq C \left( \|\nabla u\|_{L^p(\Omega)} + \|u\|_{L^r(\partial\Omega)} \right) \tag{15}$$

*In particular, if $p = r = 2$ we have that $q = \frac{2n}{n-1}$ but since $\Omega$ has finite volume, the inequality holds for each $q \leq \frac{2n}{n-1}$ and, in particular, for $q = 2$.*

**Corollary B.1.** *If $p = r = 2$, theorem B.2 holds for $q = \frac{2n}{n-1}$ and, since $\Omega$ has finite volume, it holds for each $q \leq \frac{2n}{n-1}$ and, in particular, for $q = 2$, that is*

$$\|u\|_{L^2(\Omega)} \leq C \left( \|\nabla u\|_{L^2(\Omega)} + \|u\|_{L^2(\partial\Omega)} \right) \tag{16}$$

We are now ready to prove theorem 2.2.

*Proof.* Let $\hat{u}_n$ be a sequence such that $L(\hat{u}_n, u) \leq \epsilon_n$ with $\epsilon_n \to 0$ and $v_n = \hat{u}_n - u$. From the definition of $L$ have that $\|\nabla v_n\|_{L^2(\Omega)} \leq \epsilon_n$ and similarly for $\|v_n\|_{L^2(\partial\Omega)}$, so that

$$\begin{aligned} \|v_n\|_{L^2(\Omega)} &\leq C \left( \|\nabla v_n\|_{L^2(\Omega)} + \|v_n\|_{L^2(\partial\Omega)} \right) \leq 2C\epsilon_n \to 0 \\ \|v_n\|_{W^{1,2}(\Omega)} &= \|v_n\|_{L^2(\Omega)} + \|\nabla v_n\|_{L^2(\Omega)} \leq 2(C+1)\epsilon_n \to 0 \end{aligned} \tag{17}$$

which gives us the first part of the thesis. Additionally, $\nabla v_n \to 0$. This means that the limit of $v_n$ is a.e. constant and, actually, continuously differentiable in $\Omega$. Since $v_n \to 0$ and $\lim_n \nabla v_n = \nabla v_\infty$ (limits and weak derivatives commute), at the limit $\hat{u}_\infty = u$ and, by continuity of the trace operator in $W^{1,2}(\Omega)$, we also have that $\hat{u}_\infty|_{\partial\Omega} = u|_{\partial\Omega}$ in $L^2(\partial\Omega)$. The result can be easily extended to multi-component functions by considering each component individually. $\square$

## B.3 PROOF OF THEOREM 2.3

*Proof.* We show the results for time-independent PDEs, for time-dependent PDEs it is sufficient to consider the extended domain $\tilde{\Omega} = [0, T] \times \Omega$, in which the initial (and final) conditions become boundary conditions. We also show the result for functions with one component. The result can be generalized by applying it to each component.

Let $\hat{u}$ be the trained Neural network. From theorem 2.2, we have that the network converges to the solution of the PDE $\boldsymbol{u}$ in $W^{1,2}(\Omega)$ and that the two functions coincide almost everywhere, since $\nabla(\hat{u} - u) = 0$. From this, we deduce that the PDE is satisfied by $\hat{u}$ and the boundary conditions are satisfied as well as per theorem 2.2. $\square$

**Remark B.1.** *In the case of time-dependent PDEs, we will not provide the final conditions at $t = T$, which are part of the "boundary" of the extended domain and are necessary for theorem 2.3. Since our experiments are on regular functions, this has proven not to be an issue and the neural network still converges.*

## C    ON THE USE OF EMPIRICAL DERIVATIVES

As discussed in Section 2, when analytical or true derivatives are not available, we use empirical ones via finite differences (Anderson et al., 2020) to approximate them:

$$\frac{\partial u}{\partial x_i}(\boldsymbol{x}) \simeq \frac{u(\boldsymbol{x} + h\boldsymbol{e}_i) - u(\boldsymbol{x})}{h}, \tag{18}$$

These approximations introduce errors, but the following result ensures that with a small enough $h$, these derivatives are very similar to the true ones.

**Theorem C.1.** *Let $u \in W^{1,2}(\mathbb{R}^n)$ and let $\mathrm{D}_{x_i}^\epsilon u$ be the difference quotient*

$$\mathrm{D}_{x_i}^\epsilon u(\boldsymbol{x}) = \frac{u(\boldsymbol{x} + h\boldsymbol{e}_i) - u(\boldsymbol{x})}{h} \tag{19}$$

*where $\boldsymbol{e}_i$ is the unit vector in the $x_i$ direction. Then, we have that $\|\mathrm{D}_{x_i}^\epsilon u\|_{L^2(\mathbb{R}^n)} \leq \|\frac{\partial u}{\partial x_i}\|_{L^2(\mathbb{R}^n)}$ and $\mathrm{D}_{x_i}^\epsilon u \to u_{x_i}$ in $L^2(\mathbb{R}^n)$, which means that empirical derivatives converge to weak (or true) ones as $\epsilon \to 0$. The same results holds for any open set $\Omega$, with the convergence being true on every compact subset of $\Omega$. In practical terms, empirical derivatives converge a.e. for every point distant at least $h$ from the boundary.*

*Proof.* Assuming the result for smooth integrable functions, we show the thesis using the characterization of the Sobolev space $W^{1,2}(\mathbb{R}^n)$ via $C_c^\infty(\mathbb{R}^n)$ approximations with smooth functions with compact support (Maz'ya, 2011). For each $\delta > 0$, there exists $\phi \in C_c^\infty(\mathbb{R}^n)$ such that $\|u - \phi\|_{W^{1,2}(\mathbb{R}^n)} < \delta$. First, we note that for each $u \in W^{1,2}(\mathbb{R}^n)$

$$\begin{aligned}
|u(\boldsymbol{x} + h\boldsymbol{e}_i) - u(x)| &= \left| \int_0^1 \frac{\partial u}{\partial x_i}(\boldsymbol{x} + ht\boldsymbol{e}_i) h \, \mathrm{d}t \right| \\
&\leq \int_0^1 \left| \frac{\partial u}{\partial x_i}(\boldsymbol{x} + ht\boldsymbol{e}_i) \right| |h| \, \mathrm{d}t
\end{aligned} \tag{20}$$

so that, by squaring and integrating

$$\left\| \frac{u(\boldsymbol{x} + h\boldsymbol{e}_i) - u(\boldsymbol{x})}{h} \right\|_{L^2(\mathbb{R}^n)}^2 \leq \left\| \int_0^1 \left| \frac{\partial u}{\partial x_i}(\boldsymbol{x} + ht\boldsymbol{e}_i) \right| \mathrm{d}t \right\|_{L^2(\mathbb{R}^n)}^2 \leq \left\| \frac{\partial u}{\partial x_i} \right\|_{L^2(\mathbb{R}^n)}^2, \tag{21}$$

which is the first part of the thesis. Applying this to $u - \phi$ one directly shows that

$$\|\mathrm{D}_{x_i}^\epsilon u - \mathrm{D}_{x_i}^\epsilon \phi\|_{L^2(\mathbb{R}^n)}^2 \leq \|u_{x_i} - \phi_{x_i}\|_{L^2(\mathbb{R}^n)}, \tag{22}$$

Then, we have that

$$\begin{aligned}
\|\mathrm{D}_{x_i}^\epsilon u - u_{x_i}\|_{L^2(\mathbb{R}^n)} &= \|\mathrm{D}_{x_i}^\epsilon u - \mathrm{D}_{x_i}^\epsilon \phi + \mathrm{D}_{x_i}^\epsilon \phi - \phi_{x_i} + \phi_{x_i} - u_{x_i}\|_{L^2(\mathbb{R}^n)} \\
&\leq \|\mathrm{D}_{x_i}^\epsilon u - \mathrm{D}_{x_i}^\epsilon \phi\|_{L^2(\mathbb{R}^n)} + \|\mathrm{D}_{x_i}^\epsilon \phi - \phi_{x_i}\|_{L^2(\mathbb{R}^n)} + \|\phi_{x_i} - u_{x_i}\|_{L^2(\mathbb{R}^n)} \\
&\leq \|u_{x_i} - \phi_{x_i}\|_{L^2(\mathbb{R}^n)} + \delta + \delta \\
&\leq 3\delta
\end{aligned} \tag{23}$$

where the second inequality follows from equation 22 and the last one follows from choosing a small enough $\epsilon$. The same steps are true for every open subset $\Omega$, by considering the $L_{\mathrm{loc}}^2$ norm.    $\square$

## D    EXPERIMENTAL SETUP

This Section is dedicated to additional details on the setup and tuning of the experiments.

Table 8: Computational time of 1 epoch for each experiment. For NCL we reported the computational time of 100 steps due to the definition and length of one epoch. The NCL column results are on the specific architecture of Richter-Powell et al. (2022).

| Model | Pendulum | Allen-Cahn | Continuity | Navier-Stokes | KdV | NCL |
|---|---|---|---|---|---|---|
| **PINN** | 6.0451 | 2.9809 | 6.0450 | 8.3473 | 37.197 | 9.4651 |
| **DERL** | 5.2647 | 1.2138 | 5.8674 | 4.8215 | 17.851 | 8.3582 |
| **HESL** | / | / | / | / | 25.431 | / |
| **DER+HESL** | / | / | / | / | 28.754 | / |
| **OUTL** | 3.3748 | 0.87363 | 5.3797 | 3.4959 | 14.730 | 7.8256 |
| **SOB** | 5.6974 | 1.3320 | 5.8871 | 5.2402 | 18.170 | 8.3365 |
| **SOB+HES** | / | / | / | / | 24.338 | / |

## D.1 MODEL TUNING

For each experiment, each model was tuned individually on the task to find the best parameters. This was done to extract the best performance, to have a fair comparison among the methodologies, and to measure their effectiveness in the tasks. The hyperparameters to be tuned are given by the weights of the different norms in equation 12, as well as the learning rate. The batch size was fixed for each experiment beforehand.

The tuning was conducted using the Ray library (Moritz et al., 2018) with the HyperOpt search algorithm (Bergstra et al., 2013) and the ASHA scheduler (Li et al., 2020a) for early stopping of unpromising samples. For each weight $\lambda_D, \lambda_I, \lambda_B$, the search space was the interval $[10^{-3}, 10^2]$ with log-uniform distribution, and for the learning rate, we consider discrete values in $[0.00005, 0.0001, 0.0005, 0.001]$. Most importantly, the target metric of the tuning to evaluate individual runs was the $L^2$ loss on the function $\|u - \hat{u}\|_{L^2(\Omega)}$, being the usual target in data-driven tasks.

For PDE and distillation experiments, that is all but the pendulum one, there is no train/validation/testing split, as the solution has to be learned in the whole domain. For the pendulum experiments that involve full trajectories, 30 trajectories are for training and 10 for validation at the tuning stage. These are used together as 40 training trajectories in the final model training, while 10 unseen trajectories are used for model testing. For the pendulum interpolation task in Section E.1, a total of 40 trajectories are used for in the tuning phase: points with times between $t = 3.75$ and $t = 6.25$ are unseen by the models and used for tuning/validation, while 10 new complete trajectories are used for testing.

## D.2 COMPUTATIONAL TIME

Although effective, optimization of higher-order derivatives of a Neural Network can be costly in time and computational terms. On the other hand, the functional and the Automatic Differentiation framework of Pytorch (Paszke et al., 2019) allowed us to calculate such quantities with ease and in contained time. For a complete comparison, we calculated the time to perform one training epoch for each task and model. Results are available in table 8. As expected, OUTL is the fastest having the most basic objective, while DERL beats SOB and PINN require at least one order of derivative to be calculated. It seems clear that the more terms in equation 12, the longer the training time, as seen by adding the Hessian matrices in the KdV experiment. Here, the third-order derivatives have a huge impact on PINN training. On the other hand, the approximations employed in SOB to calculate the Hessian matrices did not significantly improve the computation time compared to our approach with Automatic Differentiation. For reference, all the experiments were performed on an NVIDIA H100 GPU with 80 GB or RAM.

## E ADDITIONAL RESULTS AND EXPERIMENTAL SETUPS

In this Section, we provide additional results and details for the setup of each experiment.

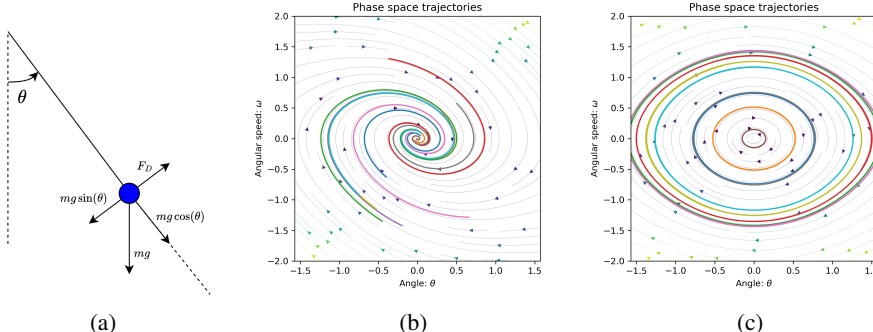

(a)            (b)            (c)

Figure 5: (a) Schematic representation of the pendulum system: the state variables are the angle $\theta$ the rope makes with the vertical direction. (b) Phase space of the damped pendulum. Arrows represent the direction and intensity of $\dot{\theta}, \dot{\omega}$. Testing trajectories are also plotted. (c) Phase space of the conservative pendulum with no dampening ($b = 0$).

### E.1   PENDULUM

We start by describing the pendulum problem from a physical point of view. We consider the rope pendulum under the force of gravity $F_g = -mg$ and a dampening force $F_d = -bl\dot{\theta}$, where $\theta$ is the angle the rope makes w.r.t. the vertical, $b$ is a parameter for the dampening force, $l, m$ are respectively the length of the rope and the mass of the pendulum. A schematic representation of the system can be found in figure 5a, and we also plot its phase space on $(\theta, \dot{\theta})$ along with some trajectories with dampening 5b or without 5c, that is the conservative case. The evolution of the pendulum is given by the ODE $ml\ddot{\theta} + mg\sin(\theta) + bl\dot{\theta} = 0$, which can be expressed in a first-order system as where $\omega$ is an alternative name for the angular speed $\dot{\theta}$, leading to equation 24 that we also presented in section 2.

### E.1.1   TUNING DETAILS

For each pendulum experiment, we sampled a total of 50 trajectories with IC $\theta_0, \omega_0$ randomly chosen in $\left[\frac{\pi}{2}, \frac{\pi}{2}\right] \times [-1.5, 1.5]$. Of these 50 trajectories, 10 are kept for testing, while the other 40 are for training and validation. For the experiments with complete trajectories, the tuning is performed with 30 as training and 10 for validation. The tuning objective was to minimize the $L^2$ loss on the validation trajectories. For the interpolation task in Section E.1.4, all 40 trajectories are used together, with the time region between 3.75 and 6.25 used for validation only and being totally unavailable during training.

### E.1.2   DIFFERENTIABILITY WITH RESPECT TO INITIAL CONDITIONS

For ODEs such as the pendulum one

$$\begin{cases} \dot{\theta} = \omega \\ \dot{\omega} = -\frac{g}{l}\sin(\theta) - \frac{b}{m}\omega, \end{cases} \tag{24}$$

the solution $(\theta(t, \theta_0, \omega_0), \omega(t, \theta_0, \omega_0))$ is actually continuously differentiable with respect to the IC $\theta_0, \omega_0$, since the right-hand sides in equation 24 are $C^1$ as well. For a proof, see Hartman (2002),

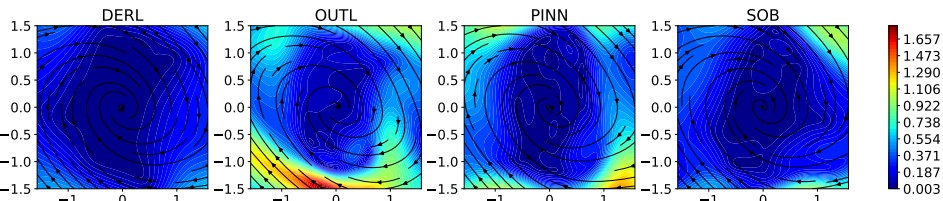

Figure 6: $L^2$ residual on the initial condition differentiability PDE. Blue is better.

Table 9: Results for the damped pendulum experiment with randomly sampled times and empirical derivatives. We report the loss on the state and its derivatives on testing trajectories, $L^2$ distances between true and predicted fields, PDE residuals, and PDE residual for the initial condition differentiability. Fields and PDE residuals are calculated at $t = 0$.

| Model | State loss | Derivative loss | Field error | PDE res. | Init PDE res. |
|---|---|---|---|---|---|
| **DERL** (ours) | 0.026976 | **0.010855** | **0.27220** | **0.21309** | **0.56515** |
| **OUTL** | 0.016530 | 0.036725 | 1.0191 | 0.83332 | 2.0075 |
| **PINN** | 0.018873 | 0.018849 | 0.55090 | 0.47905 | 1.2383 |
| **SOB** | **0.0160474** | 0.015510 | 0.45475 | 0.38576 | 1.1402 |

theorem 3.1. In this specific case, we can derive the following PDEs:

$$
\begin{cases}
\dfrac{\partial}{\partial \theta_0} \dfrac{\partial \theta}{\partial t} = \dfrac{\partial \omega}{\partial \theta_0} \\[2mm]
\dfrac{\partial}{\partial \omega_0} \dfrac{\partial \theta}{\partial t} = \dfrac{\partial \omega}{\partial \omega_0} \\[2mm]
\dfrac{\partial}{\partial \theta_0} \dfrac{\partial \omega}{\partial t} = -\dfrac{g}{l} \dfrac{\partial \sin(\theta)}{\partial \theta_0} - \dfrac{b}{m} \dfrac{\partial \omega}{\partial \theta_0}, \\[2mm]
\dfrac{\partial}{\partial \omega_0} \dfrac{\partial \omega}{\partial t} = -\dfrac{g}{l} \dfrac{\partial \sin(\theta)}{\partial \omega_0} - \dfrac{b}{m} \dfrac{\partial \omega}{\partial \omega_0}
\end{cases}
\tag{25}
$$

which are true for any trajectory $(\theta(t, \theta_0, \omega_0), \omega(t, \theta_0, \omega_0))$. In the experiments, we calculated the $L^2$ PDE residual of equation 25 in plain PINN style, to see which model learns to be differentiable w.r.t. the IC.

### E.1.3 ADDITIONAL RESULTS ON THE DAMPED PEDULUM

To complete the results of Section 4.1, we plot the $L^2$ residual on the initial condition differentiability PDE in equation 25. Figure 6 shows the domain's local $L^2$ residual. We decided to plot this instead of error differences to appreciate better the generalization power of DERL.

We repeated the same experiment using empirical derivatives, calculated by using finite difference with $h = 10^{-3}$ on the original trajectories. We also used randomly sampled times to evaluate trajectories, to simulate a setting where the time sampling is not constant. We report the relevant losses in table 9, where DERL outperforms the other methods in almost every metric, especially in those showing the generalization capability. We conclude that learning derivatives improves generalization in the models. DERL has also a simpler objective than PINNs and the other methodologies, which learn the trajectory itself as well, leading to better results.

### E.1.4 INTERPOLATION EXPERIMENT

We also performed a different experiment, where we tried to see how the models perform in an out-of-distribution setting. In particular, we chose to provide as training samples only the data simulated between $t = 0$ and $t = 3.75$, and between $t = 6.25$ and $t = 10$, effectively the $70\%$ of the trajectories' length. The task is then to correctly reconstruct the trajectory for times $t \in [3.75, 6.25]$. To have a fair comparison, we feed to the network the final state at $t = 10$, similarly to how we give

Table 10: Results for the damped pendulum experiment on trajectory interpolation. We report the loss on the unseen trajectories (New traj. loss) and the loss on the unseen part of the training trajectories (Test loss), $L^2$ distances between true and predicted fields, PDE residuals, and PDE residual for the initial condition differentiability in equation 25. Fields and PDE residuals are calculated at $t = 0$.

| Model | New traj. loss | Test loss | Field error | ODE res. | Init PDE res. |
|---|---|---|---|---|---|
| **DERL** (ours) | 0.029449 | **0.011092** | **0.33621** | **0.31579** | **0.81501** |
| **OUTL** | 0.084394 | 0.048155 | 0.67805 | 0.54420 | 1.4217 |
| **PINN** | 0.032684 | 0.015972 | 0.39573 | 0.28835 | 0.93173 |
| **SOB** | **0.019661** | 0.011633 | 0.40108 | 0.39563 | 1.1390 |

the initial one. This is because the PINN and DERL methodologies would not be able to reconstruct the final part of the trajectory without a context on that section, which SOB and OUTL naturally have by learning directly the states.

Table 10 shows the numerical results for this task. In this case, we decided to report both the error on the full new trajectories, as well as the error on the unseen section of the training trajectories. It is clear that DERL outperforms the other models in generalizing the field and the ODE to new initial conditions. Our method is the best at interpolating the training trajectories, and the second best at predicting new complete trajectories. Graphical results on the testing trajectories are available in figure 7. We conclude that OUTL is the worst at interpolating trajectories knowing the start and the end, while the others have similar performance

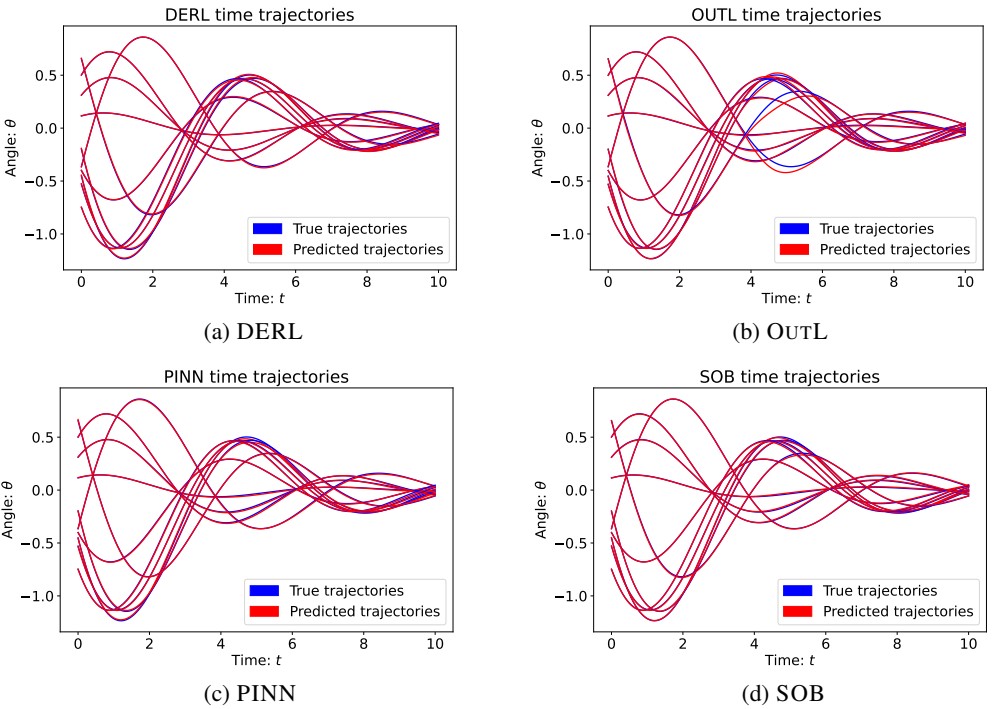

(a) DERL

(b) OUTL

(c) PINN

(d) SOB

Figure 7: True and predicted trajectories in the pendulum damped interpolation experiment.

### E.1.5 HNN AND LNN SETUP

In this Section, we describe the setup we used for the Hamiltonian Neural Network (Greydanus et al., 2019) and Lagrangian Neural Network (Cranmer et al., 2019). In particular, we will explain how we adapted these methods to learn non-conservative fields, which is not present in their original works.

Table 11: Results for the conservative pendulum experiment. We report the loss on the state and its derivatives on testing trajectories, $L^2$ distances between true and predicted fields, PDE residuals, and PDE residual for the initial condition differentiability. Fields and PDE residuals are calculated at $t = 0$.

| Model | Test loss | Test der. loss | Field error | PDE res. | Init PDE res. |
|---|---|---|---|---|---|
| **DERL** (ours) | 0.11046 | 0.073087 | **0.49334** | **0.49216** | 2.0274 |
| **OUTL** | **0.082856** | 0.10686 | 0.92589 | 0.68083 | 2.4250 |
| **PINN** | 0.22413 | 0.18899 | 0.62099 | 0.51660 | **1.8593** |
| **SOB** | 0.092996 | **0.072278** | 0.60668 | 0.66383 | 2.2725 |

In both the conservative and dampened cases, we used the setup described in their original articles without modifications in the architecture, initialization, and training. We now describe how to include the dampening effects through data or particular processing during inference.

**Hamiltonian Neural Network.** In this case, the neural network parametrizes the Hamiltonian function $H(p, q)$, from which derivatives one obtains

$$\frac{\partial p}{\partial t} = -\frac{\partial H}{\partial q}, \qquad \frac{\partial q}{\partial t} = \frac{\partial H}{\partial p}. \tag{26}$$

Equation 26 is valid in the conservative case, while in the dampened case it is sufficient to add the term $-bp$ to $\frac{\partial p}{\partial t}$, leading to the correct formulation as in equation 24. This way, the HNN learns just the conservative part of the field, which was originally made for, while we add the dampening contribution externally.

**Lagrangian Neural Network.** In this case, we need to look at the Lagrangian formulation of mechanics, in particular at the Euler-Lagrangian equation. Given a neural network that parametrizes the Lagrangian function $\mathcal{L}(\theta, \dot{\theta})$, we have that (see Cranmer et al. (2019) for the full explanation)

$$\frac{\mathrm{d}}{\mathrm{d}t}\nabla_{\dot{\theta}}\mathcal{L} - \nabla_\theta\mathcal{L} = 0. \tag{27}$$

We can add the effect of external forces, in this case, the dampening, using d'Alembert's principle of virtual work (Goldstein, 2011). In particular, given the force in vector form

$$\mathbf{F}_d = -bl\dot{\theta}[\cos(\theta), \sin(\theta)] \tag{28}$$

and the position of the pendulum as a function of the coordinates

$$\boldsymbol{x}(t) = [l\sin(\theta), -l\cos(\theta)], \tag{29}$$

the virtual work is given by

$$Q(t, \theta, \dot{\theta}) = \mathbf{F}_d \cdot \frac{\partial \boldsymbol{x}(t)}{\partial \theta} = -bl^2\dot{\theta}. \tag{30}$$

It is then sufficient to put this term on the right-hand side of equation 27 and proceed with the calculations as in Cranmer et al. (2019) to obtain the full update of the model.

### E.1.6 CONSERVATIVE PENDULUM

The setup for the conservative pendulum experiment is the same as for the dampened one, with the fundamental difference that $b = 0$ in both data generation and during training. We report the numerical results in table 11, with the same definitions as in Section 4.1. We also plot the field error differences as defined in Section 4.1 in figure 8. In this case, the results are closer in all metrics, but DERL is still the best at generalizing the field to new starting points and its error on the test trajectories is close to the best, given by OUTL, while PINN struggled.

In this case, HNN (field $L^2$ error 0.13570) and LNN (field $L^2$ error 0.085388) outperform all other methodologies as they are precisely built for this task. We still remark that their objective is solely to learn the field, which is easier than whole trajectories.

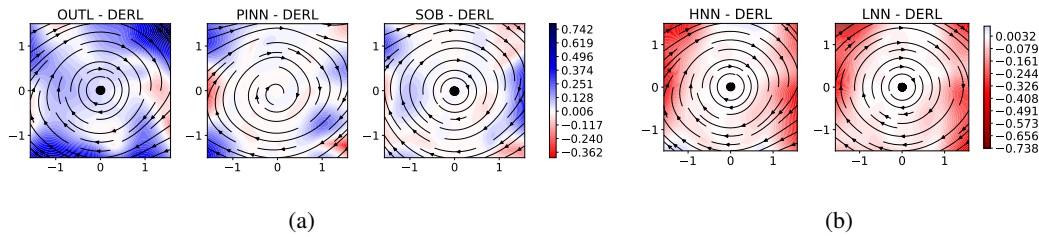

Figure 8: Comparison in the learned field at $t = 0$ for the pendulum experiment expressed as the differences of the $L^2$ errors between the other methodologies and DERL. The blue area is where we perform is better.

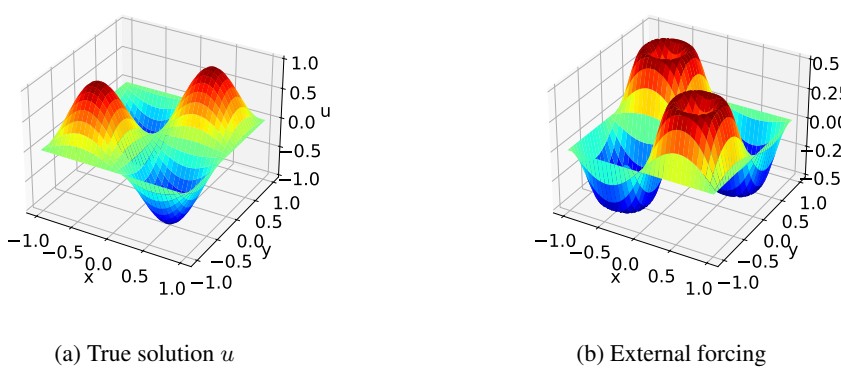

(a) True solution $u$            (b) External forcing

Figure 9: Allen-Cahn equation. (a) True solution and (b) external forcing $f$

### E.2 ALLEN-CAHN EQUATION

We start by plotting the true solution $u$ along with the forcing function $f$ in figure 9.

We now present the additional results for the Allen-Cahn equation. In particular, we show that similar results to the ones in Section 4.2 can be obtained using random points instead of an equispaced grid. For this matter, the training dataset was created by randomly sampling 10000 points in the $x, y$ plane, along with their analytical derivatives. Table 12 shows the relevant metrics, that is the $L^2$ distance w.r.t. the ground truth on $u, \nabla u$, as well as the PDE $L^2$ residual. For a graphical comparison, in figure 10 we report the $L^2$ error on $u$ and PDE residual in the form of differences between the other methodologies and DERL: blue regions are where

Table 12: Results for the Allen-Cahn equation: $L^2$ distances from the ground truth. Randomly sampled points in the domain.

| Model | $\|u - u_{\text{true}}\|_2$ | $\|\nabla u - \nabla u_{\text{true}}\|_2$ | PDE $L^2$ norm |
|---|---|---|---|
| **DERL** (ours) | **0.0090879** | 0.031058 | 0.0095793 |
| **OUTL** | 0.018933 | 0.18600 | 0.034516 |
| **PINN** | 0.014867 | 0.084936 | 0.018048 |
| **SOB** | 0.0093875 | **0.029891** | **0.0092199** |

we perform better. The key takeaway is, as in Section 4.2, that learning the derivatives is both sufficient and better than learning just $u$. We clearly see it in the results, where DERL and SOB performed best in all metrics, while PINN learning is at least 2 times worse.

### E.3 CONTINUITY EQUATION

For the experimental setup, the original solution was calculated using finite volumes (Ferziger & Peric, 2001) on a grid with $\Delta x = \Delta y = 0.01$ and time discretization of $\Delta t = 0.001$. To reduce

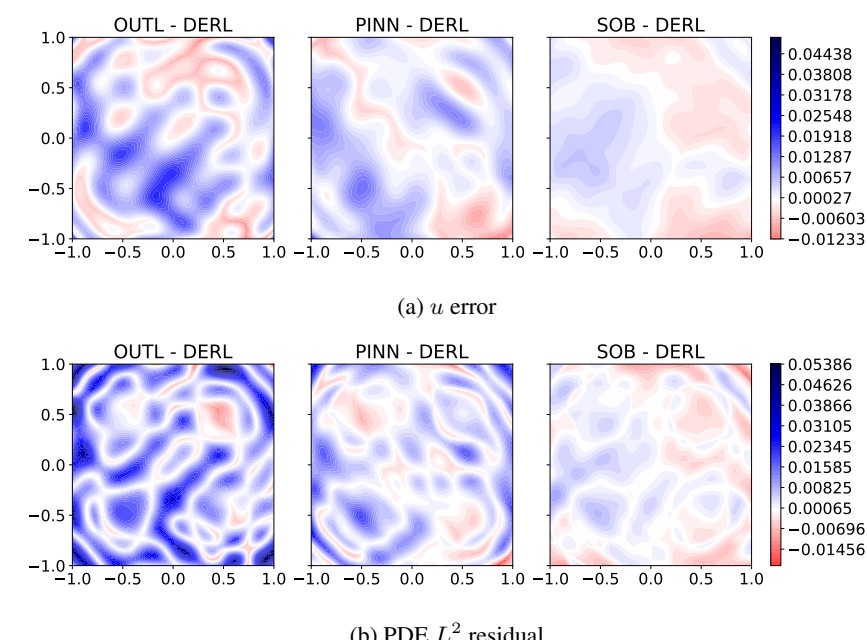

(a) $u$ error

(b) PDE $L^2$ residual

Figure 10: $L^2$ loss on $u$ and PDE residual comparison between the methodologies. Differences between the methodology and the DERL errors. Positive (blue) regions are where we perform better

the length of training we downsampled the solution to $\Delta t = 0.01$. The empirical derivatives are calculated with $h = 0.01$ on each component.

We start by showing the true solution to the continuity equation at $t = 0, 5, 9$ in figure 11a, together with the $L^2$ errors at $t = 0$ (figure 11b), $t = 5$ (figure 11c), and $t = 9$ (figure 11d). The results show clearly that the best methodologies to learn the density are DERL and OUTL, with the latter being the worst at physical consistency as reported in Section 4.3. The effect of PINN losing precision as time increases is evident, as from $t = 5$ to $t = 9$ the error grows even more.

### E.3.1 RANDOMLY SAMPLED POINTS AND EMPIRICAL DERIVATIVES

For this equation, we performed an additional experiment where we tested the effects of using an interpolated curve on the solution to randomly sample points and calculate derivatives. This simulates a situation with sparse data and no analytic derivatives of $\rho$ available. For this purpose, we used the data from the grid calculated as in Section 4.3 and used a regular grid interpolator from SciPy (Virtanen et al., 2020) with cubic interpolation for third-order accuracy. Then, we randomly sampled 10000 points in the $t, x, y$ domain, and calculated finite difference derivatives with $h = 0.001$.

Numerical results are available in table 13, while figures 12a and 12b show the comparisons on physical consistency (PDE $L^2$ residual) and the $L^2$ error on $\rho$ across methodologies. The format is the usual difference between the methods' error and the DERL one, with positive values (blue regions) meaning we are performing better. Even in this case DERL is among the best-performing models, with a $\rho$ error and physical consistency respectively simi-

Table 13: Results for the continuity equation. $L^2$ norms w.r.t. the ground truth.

| Model | $\|\rho - \rho_{\text{true}}\|_2$ | PDE $L^2$ norm |
|---|---|---|
| **DERL** (ours) | 0.027702 | 0.065162 |
| **OUTL** | 0.023268 | 0.13277 |
| **PINN** | 0.10463 | **0.049962** |
| **SOB** | **0.022879** | 0.059103 |

lar to SOB and PINN. The latter fails again at propagating the solution correctly, while OUTL is clearly the worst at learning the physics of the problem. This shows that our methodology works even with empirical derivatives calculated on an interpolation of the true solution. We remark again

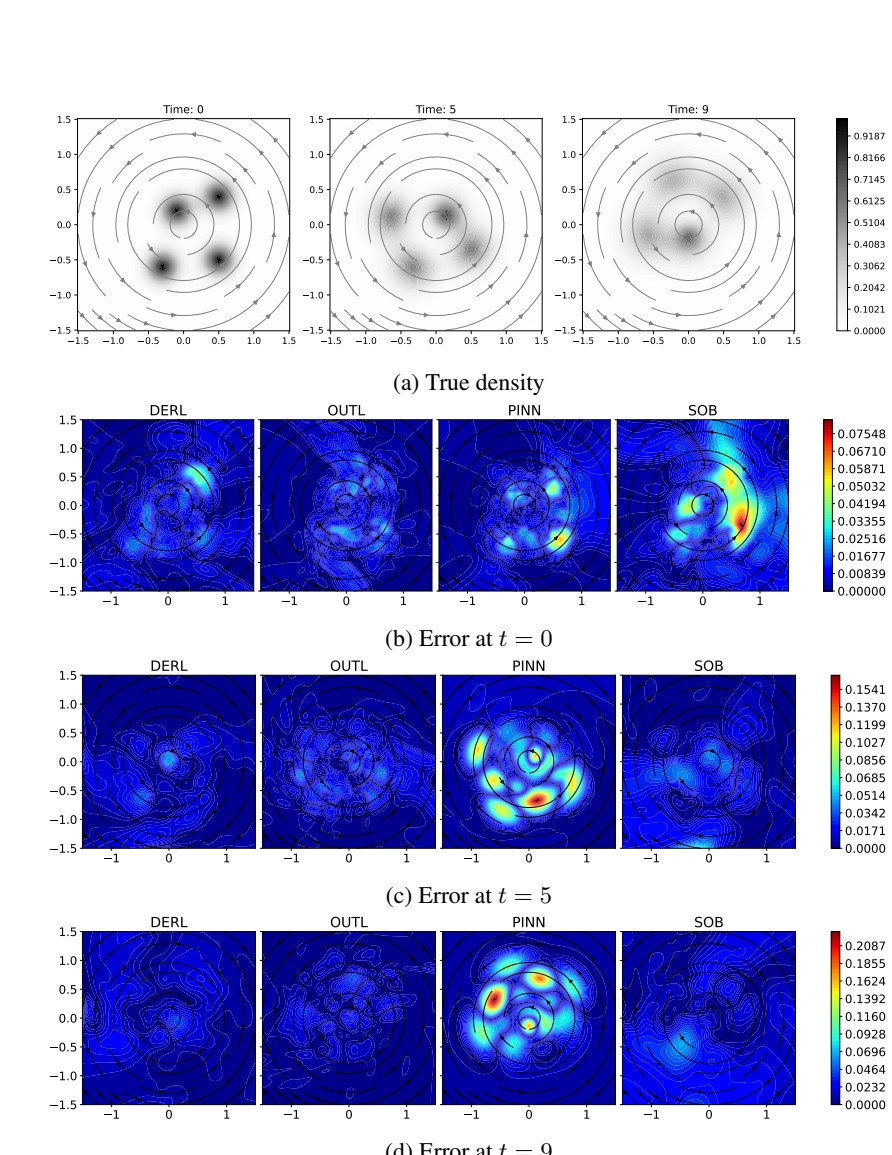

(a) True density

(b) Error at $t = 0$

(c) Error at $t = 5$

(d) Error at $t = 9$

Figure 11: Continuity equation experiment. (a) True densities at $t = 0, 5, 9$. $L^2$ errors for the compared methodologies at (b) $t = 0$, (c) $t = 5$, (d) $t = 9$.

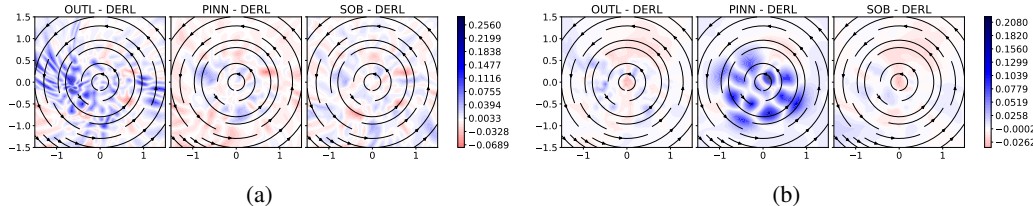

(a)                                              (b)

Figure 12: Continuity equation task with randomly sampled points: (a) Comparison of $L^2$ PDE residual on the domain at $t = 5$. Differences between the other methods' residuals and DERL. Blue regions are where we perform better (b) Same plot but with $L^2$ distance w.r.t. the true solution $\rho_{\text{true}}$

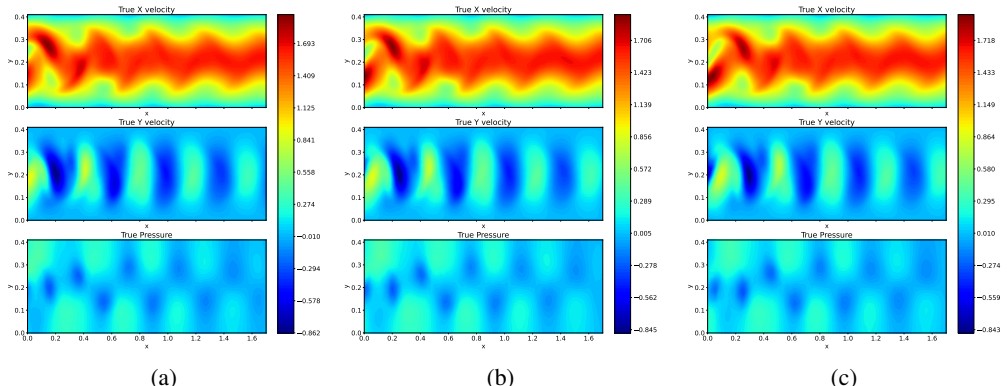

(a)                             (b)                             (c)

Figure 13: Navier Stokes equations true solution (a) $t = 0$ (b) $t = 1$ (c) $t = 2$

the importance of learning the derivatives, as DERL and SOB are the best all-around models for this task.

### E.4  NAVIER-STOKES EQUATIONS

In the following Sections, we describe the original setup for the Navier-Stokes experiment, as well as provide additional results.

### E.4.1  SETUP

The full domain is the rectangle $[0, 2.2] \times [0, 0.41]$ with a circular obstacle of center $[0.2, 0.2]$ and radius $\frac{1}{20}$. The IC is $\boldsymbol{u} = \boldsymbol{0}, p = 0$ everywhere, while BC are given by: $\boldsymbol{u} = 0$ on the top, bottom, and obstacle boundary, $\boldsymbol{u}_2 = 0$ on the left boundary and $p = 0$ on the right one. The horizontal speed $\boldsymbol{u}_1$ at the left boundary is given by the function

$$b(t, y) = \frac{y(0.41 - y)}{0.41^2} \frac{e^{5t}}{6e^{20} + e^{5t}} \tag{31}$$

The equation was solved for $t \leq 10$ with the finite volumes method and time discretization (Anderson et al., 2020) with second-order polynomials and a grid with maximum size 0.0005 using the software Mathematica (Inc., 2022). In the learning task, we instead considered only the region at the right of the circular obstacle starting at $x = 0.5$, similar to Raissi et al. (2019). We focused on times $8 \leq t \leq 10$, where the solution is periodically stable. See figure 13 for the plots of $\boldsymbol{u}, p$ at those times. The IC is obtained from the reference solution at $t = 8$ and the BC is changed accordingly, particularly for the new left boundary $x = 0.5$. The data, with partial derivatives from Mathematica, is then saved with a grid size of $\Delta x = \Delta y = \Delta t = 0.01$. The MLPs take as input $(t, x, y)$ and predict $\boldsymbol{u}(t, x, y), p(t, x, y)$ and are made of 8 layers of 128 units each with Tanh activation and were trained for 200 epochs with batch size 512. While in Raissi et al. (2019) a latent potential is modeled to satisfy automatically $\nabla \cdot \boldsymbol{u} = 0$, here we model the components of the velocity independently, to see if we can learn to satisfy 2 PDEs at the same time.

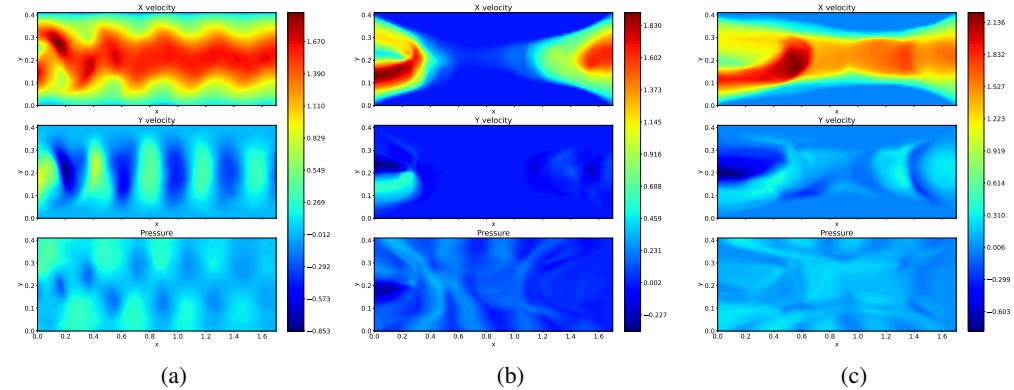

Figure 14: Results from the trained PINN model for the Navier Stokes experiment at (a) $t = 0$ (b) $t = 1$ (c) $t = 2$

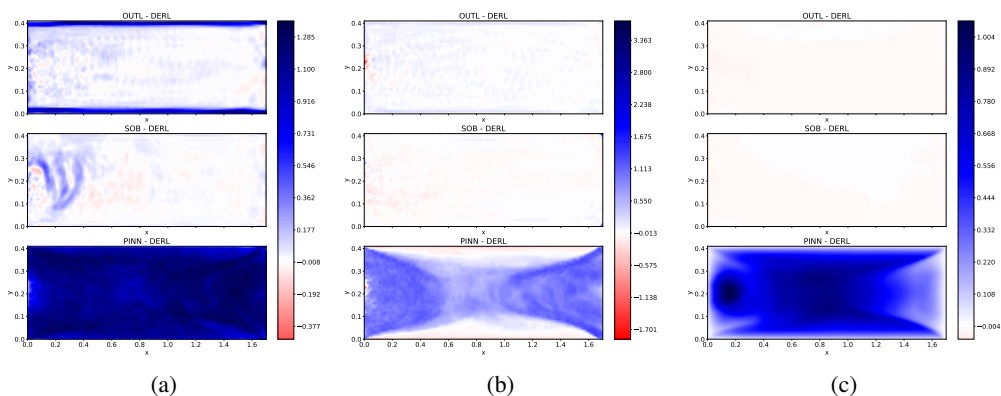

Figure 15: Navier Stokes equations results. Error difference between DERL and other methods for (a) the momentum equation residual (time-averaged), (b) the incompressibility equation residual (time-averaged), and (c) for the learned solution. Blue regions are those where DERL performs better. Results for OUTL and PINN had to be capped at $1.5$.

### E.4.2 ADDITIONAL RESULTS FOR SECTION 4.4

We report here additional results and the plots relative to the Navier-Stokes experiment with true derivatives and points from the original grid.

As said in Section 4.4, the PINN was able to learn only the IC at $t = 0$, failing to propagate it correctly for higher times and diverging from the true solution by a large margin. This can be seen from figure 14 where we plot the results to be compared with figure 13. We then plot the error difference between DERL and the other methodologies in the learned solution (figure 15c) and the PDE residual of both the incompressibility (figure 15b) and momentum equation (figure 15a). The comparison is expressed in terms of the time-averaged difference between the method's $L^2$ local PDE residual and the one from DERL so that blue regions are where we perform better. We remark that OUTL and PINN had to be capped at $1.5$ for the sake of image quality, as they performed very badly in some regions. The figures clearly show the great performance of DERL in learning to be consistent with the momentum equation. OUTL, on the other hand, struggles especially near the boundary.

### E.4.3 RESULTS ON RANDOMLY SAMPLED POINTS

The results tell us a very similar story to the one in Section 4.4, even with empirical derivatives and interpolated data. Derivatives are sufficient to learn the overall solution and provide more physical information than the values of $\boldsymbol{u}, p$ only. Ex-

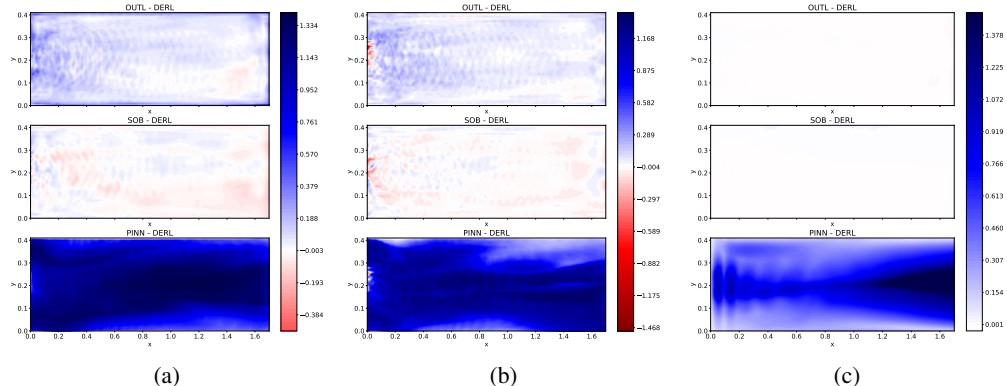

(a)        (b)        (c)

Figure 16: Navier Stokes equations results for randomly sampled data and empirical derivatives (a) Comparison between DERL and other methods for the momentum equation residual (time-averaged). Results for OUTL had to be capped at $1.5$ for image quality reasons. (b) Comparison for the incompressibility equation residual (time-averaged). (c) Comparison for the $L^2$ error on the solution (time-averaged). Blue regions are those where DERL performs better.

cluding PINN, the error on the solution is similar across methods but the momentum equation consistency of learning with derivative information is unmatched by OUTL.

Table 14: Results for the Navier Stokes experiments on randomly sampled points. $L^2$ error on the final solution, $L^2$ norms of the residuals of the 2 PDEs. Norms are calculated across time-space.

| Model | $L^2$ **error** | **(E4.M)** $L^2$ **norm** | **(E4.I)** $L^2$ **norm** |
|---|---|---|---|
| **DERL** (ours) | 0.020290 | 0.33351 | 0.35802 |
| **OUTL** | **0.017841** | 0.61518 | 0.27716 |
| **PINN** | 0.81586 | 13.115 | 7.8778 |
| **SOBL** | 0.021293 | **0.32908** | **0.25817** |

Similarly to the continuity experiment, we tried a setting with randomly sampled points in the domain, obtained from a third-order interpolation of the true solution with SciPy. Empirical derivatives are obtained via finite differences with a displacement of $h = 10^{-3}$. Numerical results are reported in 14, while the time-averaged comparison in $L^2$ error and physical consistencies are available in figure 16.

### E.5 KNOWLEDGE DISTILLATON

#### E.5.1 KORTEWEG-DE VRIES EQUATION

**Teacher and student model setup.** The reference solution is first obtained using the Scipy (Virtanen et al., 2020) solver with Fast Fourier Transforms on a grid with $\Delta x = \Delta t = 0.005$. Then, a PINN is carefully trained on equation (E5) in table 1 and is treated as the teacher model to distill knowledge from. For the teacher model, we used an MLP with 9 layers of 50 units, $\tanh$ activation, and a batch size of $64$. We trained the network for 200 epochs with hyperparameters given by the tuning process with minimum $u$ loss as the objective. After the teacher's training, we save a dataset of its outputs $u_{\text{PINN}}(x, t)$, its derivatives $\nabla u_{\text{PINN}}(x, t)$ and its hessian matrix $\boldsymbol{H}_{\text{PINN}}(x, t)$, evaluated at the same points it was trained on and calculated via automatic differentiation (Baydin et al., 2018). Then, the student networks are trained using these datasets and the original BC and IC.

**Additional results.** We report here additional plots related to the Korteweg-de Vries experiment in Section 4.5.1. Figures 17a and 17b show on the $(t, x)$ domain the comparison of the errors on $u(t, x)$ and the local PDE residual for the tested distillation methods.

As we can see, every tested method learns similarly to approximate the true solution $u(t, x)$, with slightly better performances on models with fewer derivatives to learn. On the other hand, more derivatives help the consistency of the model to the underlying PDE compared to no derivatives, as

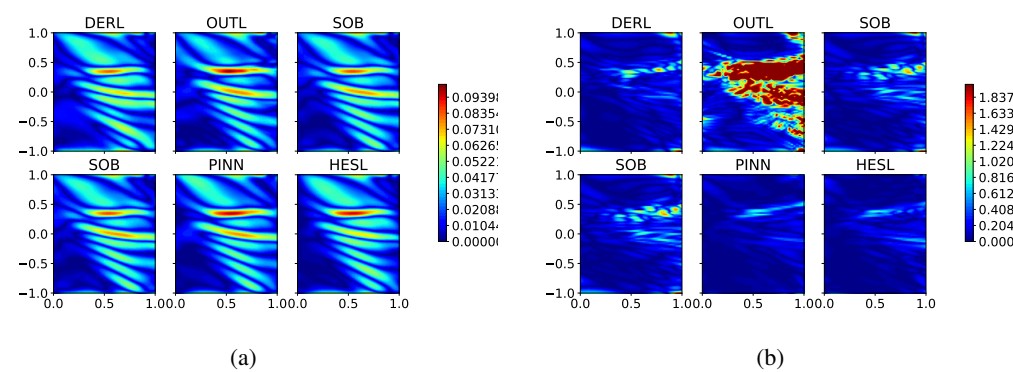

(a)  (b)

Figure 17: Korteweg-de Vries equation experiment. (a) Errors with respect to the true solution $u_{\text{true}}$ in the $(t, x)$ space. (b) Local PDE $L^2$ residual, physical consistency. In OUTL, the residual is capped at 2 for clarity reasons on the other plots.

in OUTL, which fails in this regard. It is worth noticing, as reported in Section 4.5.1, that the PINN was not able to optimize the BC, while other methodologies adapted well to them.

### E.5.2 NCL DISTILLATION

**Model setup.** The model architecture is the same as in Richter-Powell et al. (2022), that is a MLP with 4 layers of 128 units, trained for 10000 steps on batches of 1000 random points in the 3D unit ball. As in the previous task, each student model has the same architecture as the teacher model, and the hyperparameters are tuned for the best loss on imitating the teacher's output. Each student model is trained for 10 epochs on the dataset created by the output and derivatives of the teacher network. We used more than 1 epoch to ensure convergence of the models. We remark that the errors on the solution are calculated with respect to the NCL model, as no numerical solver we tried converged to the true solution.

**Additional results.** We report here additional plots and results for the distillation experiment on the Neural Conservation Laws model (Richter-Powell et al., 2022). Figure 18 shows the comparison among methodologies in terms of error differences between OUTL, SOB, and DERL so that positive blue regions are where our methodology performs better. We plot the errors on $u, \mathrm{D}u$ with respect to the NCL reference model on the $Z = 0$ plane, as well as the PDE residual error on the momentum (E6.M) and incompressibility (E6.I) equations in table 1. Except for some initial conditions, DERL outperforms the other two methodologies, especially in the $\boldsymbol{u}, \mathrm{D}\boldsymbol{u}$ errors and in the momentum equation consistency.

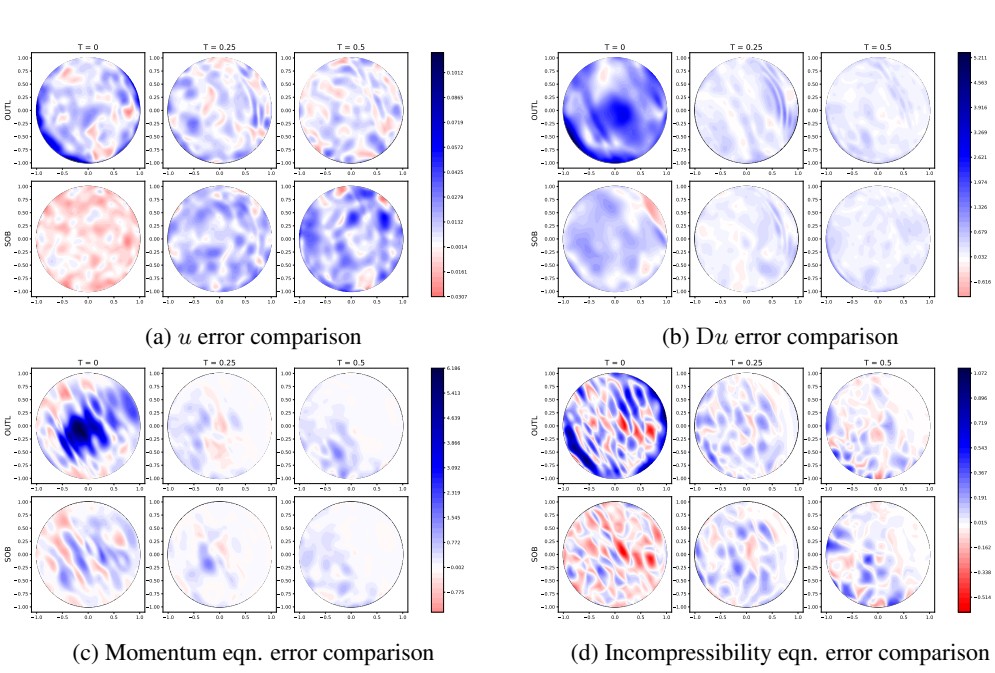

(a) $u$ error comparison

(b) $\mathrm{D}u$ error comparison

(c) Momentum eqn. error comparison

(d) Incompressibility eqn. error comparison

Figure 18: Neural Conservation Laws distillation. Comparisons between the methodologies on $u, \mathrm{D}u$ errors w.r.t. the reference model and PDE residuals on the $Z = 0$ plane. All plots represent the difference between the method error and the DERL error, so that positive (blue) regions are where DERL performs better.

