# OpenReview forum: "Derivatives Are All You Need For Learning Physical Models"
_ICLR.cc/2025/Conference — Submitted to ICLR 2025_

### Official Review · Reviewer_qf9B · 2024-10-25

**Soundness:** 1
**Presentation:** 2
**Contribution:** 1
**Rating:** 1
**Confidence:** 3

**Summary:**

The paper proposes training neural networks by fitting its derivatives to the derivatives of a data set. The authors evaluate this method on physical systems, where the derivatives are obtained from the solution trajectories of differential equations.

I see several fundamental issues with this work:
1) The method is presented as ‘physically consistent’, like physics informed neural networks (PINNs), because it uses partial derivatives. Later, it is argued that the authors’ method is easier to train because the network’s partial derivatives are compared to ‘individual targets instead of being entangled together’. The loss of PINNs incorporates the relationship between partial derivatives; this is what makes them physically consistent. To me, the proposed method appears to be a supervised method through and through.
2) The method is presented as a solution method to physical systems (see for instance in the conclusion: ‘We showed theoretically and experimentally that our method successfully learns the solution to a problem and remains consistent with its physical constraints, …’). What was actually done in the experiments is that the solution was known beforehand, either determined by classical numerical methods or it was a system with a known analytical solution. Then a neural network was fitted to that solution. While it is true that a solution was learned in the literal sense, the statement is misleading, as it indicates their method is a solution method for differential equations, just like PINNs. But more importantly, it raises the question what the method’s intended purpose then is.
3) The role of physics in this method is not really clear. As explained, the proposed technique is a method to fit a neural network to a given curve, not a solution technique. The authors argue that this is a consequence of the uniqueness theorem from the theory of ordinary differential equations, where dx/dt = f(x,t). What the method is actually based on is a simplified case, where the derivatives are known beforehand, corresponding to dx/dt = f(t) and within the scope of the well-known fundamental theorem of calculus. That a curve can be reconstructed from its derivatives is not a specific property of dynamical systems but only a property of a differentiable curve. For this reason, I would also recommend to present this work more related to Sobolev training than to physical systems and PINNs. The connection to Sobolev training should be clear from the beginning due to their strong similarities.
4) In section 2.1, theoretical analysis, the statements of the mathematical theorems are hard to follow as some of the quantities are not declared, or used in a way that does not make sense. For instance, it is not clear what the arrow in theorem 2.1 precisely means as there are no sequences that could somehow be connected to convergence of some sort. Consequentially, the proof also makes no sense to me.

**Strengths:**

See above.

**Weaknesses:**

See above

**Questions:**

See above.

---

> ### Author Response · Authors · 2024-11-14
> **Author rebuttal**
>
> Dear reviewer,
>
> to answer the several fundamental issues raised:
>
> 1. our method is a supervised method through and through, and we never claimed the opposite. The key insight of our work is indeed that with a supervised method we are able to obtain the same physical consistency of a PINN, if not to improve it. The other tested methods underperform compared to DERL and most works on supervised learning of physical consistency tend to avoid this topic. This supervised approach also provides some advantages, such as theoretical guarantees of convergence to the true solution (which are still in debate for PINNs in general) and more efficiency during training, as proven by the computational times in Table 8. As pointed out by your review, the relationship between partial derivatives is what makes a model consistent and we stress again that DERL can achieve SOTA performance while using separate signals for each derivative, which is novel and remarkable.
>
> 2. The solution was known beforehand by us, but it is still unknown to the model. The full solution is calculated to build the task and the data used to train the model, like any supervised learning approach. We intend to learn a solution to the PDE, just like PINNs, but in a supervised manner, while comparing to PINNs to see how our model performs compared to a SOTA model for physical consistency. Additionally, we believe the review completely avoids acknowledging our contribution to transferring physical knowledge and constraints between models, which is in our opinion a novel contribution. The results are encouraging both in terms of transferring successfully physical constraints, as well as improving the teacher model on others.
>
> 3. As pointed out in section 2 of our work, we exploit the fact that “a curve can be reconstructed from its derivatives” to learn a solution to a PDE, showing the advantages in terms of physical consistency. We fit a neural network to a curve just like any deep learning solution, from MLPs to LLMs. Furthermore, physical consistency is a key issue in these topics, and that is why we focused on problems described by ODEs/PDEs. Just like PINNs, our model learns to fulfill an equation but it does so without direct access to it.
>
> 4. We will try to improve the readability of the section in the next version. However, we want to stress that all the mathematical objects that are present in the theoretical statements and proofs are defined in section 2, such as the loss $L(u,\hat{u})$, the learned function $u$, and its approximation $\hat{u}$. Lastly, we want to remind the reviewer that the definition of a limit (or convergence) can be given even without an explicit sequence, as any textbook on calculus would show.

---

> > ### Comment · Reviewer_qf9B · 2024-11-27
> >
> > Dear Authors,
> >
> > Thank you for your response.
> >
> > Besides supervised training, physics-informed training is a another way to optimize neural networks that offers several advantages, which I believe are grouped together under the term ‘physical consistency’ in your paper.
> >
> > The proposed method is often described as physically consistent, which led me to think you considered your method to be some sort of physics-informed training. It is true, you never classified it explicitly as physics-informed, but also not as supervised. Often, formulations are vague, which makes it difficult to pinpoint the issue.
> >
> > But then, the question is what makes your method physically consistent if it’s not based on physics-informed training? This term ‘physically consistent’ remains abstract despite being part of your central claim, roughly ‘physically consistency without PDE loss’. It should really be specified. As you correctly cited, physics-informed losses are more difficult to optimize. People are interested in these harder loss functions in situations where there are issues in the data set for a supervised setup. For instance, a PINN is not constrained by the discretization errors of the method that generates the data for a given PDE. Or on inverse problems with multiple solution modes, supervised method typically interpolate between individual solutions, whereas physics-informed methods fit one specific solution mode. I don’t see such notions of physical consistency demonstrated in this work and would not expect the proposed method to achieve that since it is ultimately a supervised method.

---

### Official Review · Reviewer_rhs6 · 2024-10-31

**Soundness:** 2
**Presentation:** 2
**Contribution:** 2
**Rating:** 3
**Confidence:** 4

**Summary:**

The paper primarily addresses a critical question in the training of neural PDE solvers: what should the learning objective be, or, in other words, how should the loss function be defined. Previous work has commonly utilized either the discrepancy between predicted and true state values or the residual form of the PDE itself as the loss. In contrast, this paper proposes a new loss function based on the derivatives of the state with respect to the time variable \(t\) and spatial variable \(x\) as supervisory signals. The authors also theoretically demonstrate that gradient-based supervision alone is sufficient and necessary for convergence toward the target solution. Extensive numerical experiments are conducted to validate the proposed method’s effectiveness.

**[-]** The main content of the paper centers around introducing the authors' perspective and framework—specifically, the IMPORTANCE OF DERIVATIVE LEARNING. However, it lacks a detailed discussion of challenges encountered during this process and the proposed solutions. Thus, the primary contribution lies in the perspective itself, asserting that derivative-based supervisory signals are both important and novel, which is the key selling point of the paper. However, from my viewpoint, this perspective is not entirely novel, as related approaches have been explored in previous work. Here are a few examples:

- [1] Sitzmann, Vincent, et al. "Implicit neural representations with periodic activation functions." Advances in neural information processing systems 33 (2020): 7462-7473.
    - Research has shown that first- or second-order gradients alone can be effective for image supervision and has examined the associated challenges, such as activation function selection.
- [2] Li, Chongchong, et al. "Gradient information matters in policy optimization by back-propagating through model." International Conference on Learning Representations. 2022.
  -  In reinforcement learning model training (closely related to the ODE scenarios in this paper), gradient information has been shown to be crucial for downstream control tasks, with solutions also proposed to address this.
- [3] D'Oro, Pierluca, and Wojciech Jaśkowski. "How to learn a useful critic? model-based action-gradient-estimator policy optimization." Advances in Neural Information Processing Systems 33 (2020): 313-324.
  -  In the learning of value functions in reinforcement learning, since downstream policy learning relies on the gradient of the value function (similar to how PDE or ODE gradients are essential for model evolution in this paper), gradient-based supervision is incorporated into value function learning.

These are merely indicative examples, suggesting that this topic has been explored in deep learning literature. The authors should demonstrate awareness of such research to avoid reinventing the wheel.

**[-]** Moving to the specific research content, Definition 2.1 attempts to prove the sufficiency of the proposed loss, but it does not clarify whether methods like PINN, OTL, or SOB are also sufficient, nor does it establish any clear superiority of the proposed loss over these methods.

**[-]** If we are given a PDE with an analytical form along with initial and boundary conditions, how would the proposed method obtain supervisory signals? Would it still require numerical methods to generate data \(u(x, y, t)\), followed by gradient estimation via interpolation?

**[-]** For a system with a 3D input (x, y, t) and a 2D output (u, v), first-order gradient information actually forms a Jacobian matrix (2x3). Would the second-order Hessian matrix then be a higher-dimensional matrix? What computational resources would be required to estimate such derivative information?

**[-]** Regarding the experiments, could the authors clarify why the metrics in Tables 2, 3, 4, 5, and 6 differ so greatly ( Perhaps  Table 3 includes the three more intuitively understandable metrics)?

**[-]** The proposed method does not perform well in Tables 4 and 5, and its performance in Table 7 is comparable to that of SOB (which aligns with my earlier concerns about the theoretical part). I reasonably suspect that, if given appropriate weight adjustments, the SOB method would be at least as effective as the proposed loss, given that the proposed loss is a subset of the SOB method’s loss function. The paper provides no evidence (or intuitive argument) to suggest that adding a reasonable constraint to a loss would have adverse effects.

**[-]** The paper's presentation could be improved. Too many important details are placed in the appendix, leaving the main text with almost no information about the algorithm and experimental specifics—such as how the loss function is implemented. There is nearly one page of unused space in the main text, which could be used to provide these crucial details.

**Strengths:**

see above

**Weaknesses:**

see above

**Questions:**

see above

---

> ### Author Response · Authors · 2024-11-14
> **Author rebuttal**
>
> Dear reviewer,
>
> we would like to begin by noting again that our work is not limited to learning dynamical systems and PDEs, as the review summary suggests. Instead, we also mention how our method can be used to transfer physical constraints between models, as we described again in the general comment.
>
> Moving to the specific points:
>
> 1. We would like to mention that technical challenges were not mentioned due to the nature and limited space of a conference paper like this one (which according to the provided guidelines in 9 pages). Instead, we focused on the main message of our work, which is the importance of derivatives for the physical consistency of a model. We thank the reviewer for the references, but we stress that none of these examples is applied to physical systems per se, nor do they discuss the physical consistency of the model.
>
> 2. About whether OUTL and SOB  are sufficient to learn the function, the answer is yes due to their loss, which directly contains a term for learning u. About PINNs, the ability to learn $u$ solely from the PDE, the initial and the boundary conditions is still up to debate as examples show PINN can deviate from the true solution (see [1], [2] and our work for the continuity equation experiment in section 4.3). Theoretically, there is no guarantee that PINNs work for all kinds of PDEs.
>
> 3. The specific way to obtain the numerical signals for u(x,y,t) depends on the specific experiment. In our work, we explained how this was obtained in each case. In some cases, they were calculated directly from the analytical solutions, in other cases they were obtained from a numerical method. Finally, DERL worked even using empirical derivatives starting from grid values of u or an interpolation of it.
>
> 3. Yes, the Jacobian and Hessian matrices scale up as noted, requiring an incremental use of resources. Nevertheless, the computational times required to perform such operations (available in table 8) are similar to the ones of OUTL, that is supervised learning of u, lower than SOB, and much lower compared to PINNs. Therefore, computational resources are not an issue for this model realistically.
>
> 4. The different metrics in the tables are related to the different nature of the experiments. We will try to make them more unified where possible.
>
> 5. We would like to remind the reviewer that extending Sobolev learning with more hyperparameters and terms was out of the scope of our work, which actually aims to send a different message: the importance of learning the derivatives and their sufficiency, as well as the ability of DERL to transfer physical constraints.
>
> 6. Finally, we would like to remind the reviewer that the official author guidelines from ICLR suggest writing the paper within 9 pages, to allow for additions in the final versions. We complied with these limits provided by the organizers and we believe it would be unfair to penalize our work for having followed the guidelines.
>
> [1] Sun, Luning et al. “Surrogate modeling for fluid flows based on physics-constrained deep learning without simulation data.” Computer Methods in Applied Mechanics and Engineering (2019)
>
> [2] Wang, Sifan et al. “Understanding and Mitigating Gradient Flow Pathologies in Physics-Informed Neural Networks.” SIAM J. Sci. Comput. 43 (2021): A3055-A3081.

---

### Official Review · Reviewer_vHKa · 2024-11-03

**Soundness:** 2
**Presentation:** 2
**Contribution:** 2
**Rating:** 3
**Confidence:** 4

**Summary:**

This paper proposes a method for learning solutions to ODEs/PDEs by training with information of the derivatives of the solution, called DERL. The method using a loss term consisting of the error in the derivatives and the error in the initial/boundary conditions. DERL is compared to PINN, OUTL, and SOB on several differential equation problems. Additional, the method is applied to learning the solution from a "teacher" model.

**Strengths:**

The theorems are sound, other than some notational issues. The experiments are extensive and emperically, the proposed model performs best in most cases.

**Weaknesses:**

The biggest concern is that the comparison between DERL and PINN is unfair because PINN does not have access to any information about the true solution within the domain. It only has access to the initial/boundary data and the PDE. DERL on the other hand has access to information about the true solution. It is access the derivative of the solution rather than the solution itself, but the PINN does not have any information about the true solution within the domain. So the results of DERL learning better than PINN are predictable.
Another weakness is that all of the comparisons are against other methods that are not the most recent. Most of the other methods are from 2019 or later. It would be good to see comparisons to more recent methods.

**Questions:**

* In Table 3, the PINN loss is listed as 030950. Is this supposed to be 0.030950?
* For the PINN distillation, you list the model architecture for the teacher model. Is it the same architecture for the student model? Or is the student model a smaller model?
* You have figures comparing DERL to HNN and LNN. Do you have loss infomation for HNN and LNN similar to Table 2 to compare?

---

> ### Author Response · Authors · 2024-11-14
> **Author rebuttal**
>
> Dear reviewer,
>
> we would like to emphasize that, while DERL has access to information on the true solution of the problem for the training datapoints, all supervised learning methods for physical systems do. Unlike DERL however, other supervised methods fail to learn the physical consistency of the solution and drastically underperform PINNs. This can be seen in part by the results of OUTL, which knows perfectly $u$ but fails in the PDE residual loss.
>
>
> It is not obvious that learning the partial derivatives automatically leads to a high physical consistency. In this sense, PINNs have an advantage since they know and learn the full description of the problem in terms of the PDE. DERL, like other supervised learning methods, does not but it is still capable of performing comparable or even better than PINN.
> Finally, when true analytical derivatives are not available, empirical derivatives can be used. To this end, in the paper we performed experiments with partial derivatives calculated empirically based solely on $u$ while still obtaining state-of-the-art performance on physical consistency.
>
> Coming down to the questions:
>
> 1. Yes, that is a typo, the correct loss is 0.030950. Thanks for spotting this.
>
> 2. The architecture of the student model is the same as the teacher one, as briefly described at the beginning of sections 4.5.1 and 4.5.2. We will make it more evident in a subsequent version of the paper.
>
> 3. Yes, we can produce these values and put it in the paper, although this deviates from the true message of this part of the work. HNN and LNN will outperform all the other methods in most metrics since they do not predict entire trajectories but rely on external solvers. Instead, what we most cared about was the performance on generalizing to new initial conditions, at which DERL outperforms even these SOTA methods for the pendulum dynamics.

---

> > ### Comment · Reviewer_vHKa · 2024-12-02
> >
> > Thank you for your response. My concern remains that DERL and PINN are supervised and unsupervised respectively and so it is an unequal comparison. For comparing DERL to OUTL, while both are supervised, DERL has access to more data (the derivatives) then OUTL, so again it is an unequal comparison. For this reason, I maintain my score.

---

### Official Review · Reviewer_xbjs · 2024-11-04

**Soundness:** 3
**Presentation:** 2
**Contribution:** 1
**Rating:** 3
**Confidence:** 4

**Summary:**

This paper shows that it is possible to "learn" the solution to a particular PDE from the boundary/initial conditions as well as labeled data for the partial derivatives at collocations points in the space-time domain. In essence, this is an alternative approach to supervised learning of the solution, but using the gradients of the solution field rather than the solution itself.

**Strengths:**

The paper is well-presented and structured.

**Weaknesses:**

1. Although this paper is technically sound, my first major concern is that it is misleading in terms of what the proposed method achieves. For example, sentences such as "We claim that it is possible to learn physically consistent models without explicit knowledge about the underlying equations" or "DERL outperforms PINNs and other state-of-the-art approaches" make it sound like the paper proposes a novel method to solve PDEs, while that is NOT the case at all. What this paper proposes is a supervised learning approach for a particular PDE, where instead of training the network on the values of the solution, the network is instead trained on the gradients of the solution.
2. I fail to see the utility of this method in practice: it can't be used to calculate a PDE solution without a priori knowledge on the gradients of the solution. Thus, this paper's real contribution seems rather trivial to me, and it would be more suitable for a workshop rather than ICLR.

**Questions:**

It is surprising that the proposed method outperforms Sobolev learning, which trains the network using both the solution itself as well as its gradients. It seems that more data should make learning easier, especially when we have labeled data for the direct outputs of the neural network. Can you explain why you are getting such results?

---

> ### Author Response · Authors · 2024-11-14
> **Author rebuttal**
>
> Dear reviewer,
>
> 1. As described in the general comment, our main focus was to learn physical systems starting from available data in a supervised way and never claimed the opposite. In particular, we focused on improving the physical consistency of the model, which was compared to a SOTA model such as the PINN with its own PDE consistency metric, showing better performance in general. First, we stress that our main message is novel: derivatives are sufficient to learn the physical constraints of a system with PINN-like performance and we are the first to show that both theoretically and experimentally. Second, we would like to emphasize again one part of our work that was not acknowledged in the review: the ability to transfer such constraints from a trained model to a student one. We are the first to employ derivative distillation to perform this task, achieving an excellent physical consistency and improving other constraints of the teacher model. We would like our work to be evaluated for its full potential, while this review seems to ignore this part entirely.
>
> 2. To better understand our point and work it is necessary to shift the focus away from learning a system given the definition alone. In these terms, there are plenty of data-driven supervised methods to learn physics in literature, with many recent examples that are published every year. We described many of them in the related works section and reported a few in our general comments. Many of these methods learn the solution directly from its data and lack physical consistency, which was the key achievement for DERL: an improved physical consistency, similar to PINNs, while learning the solution as well with great performance. Furthermore, DERL works even when the analytical derivative is not available, using an empirical derivative calculated from $u$ or an interpolation of it. Lastly, we do not believe our contribution to be “trivial”, especially since, to the best of our knowledge, we are the first to employ and consider physical knowledge transfer between models. This would be of great impact in research, allowing for an incremental definition of physical models that can satisfy multiple constraints, increasing efficiency and learnability, as proven by our results in section 4.5.1, where distilling the physical knowledge allowed for better performance on boundary conditions. Of course, we remain available for further discussion in case the reviewer is aware of similar approaches in the literature.
>
>
> Concerning the question of why Sobolev learning underperforms our method, we attribute this effect mainly to two facts:
>
> 1. Loss gradients produced by Sobolev learning contain information about both the function $u$ and its derivative, leading to a gradient direction that is less focused on physical consistency such as the one from DERL.
>
> 2. Furthermore, learning directly $u$ can deteriorate the performance of the model in learning the physical constraints in terms of PDE residual loss. OUTL is an extreme case for this since it only learns u and fails in most cases to learn the physics of the problem.
>
> Therefore, we attribute the worse performance to a combination of these two effects: a more complex gradient direction, which pushes the model to mediate between derivatives and $u$, and too much information on $u$ directly. Lastly, we stress that this analysis of the effects of the different loss terms for the physical consistency of the model is completely novel.

---

### Author Response · Authors · 2024-11-14
**Overall rebuttal by Authors**

We thank the reviewers for their comments and suggestions and hope to engage in a fruitful discussion during this next phase. Here we provide answers to the most important points raised by the reviewers. We also reply to each reviewer individually.

First, we would like to point out that the transfer of physical knowledge between models through derivative distillation, one of the key contributions of our work, was largely disregarded in the reviews. We highlighted the novelty and relevance of derivative distillation in the introduction and methodology. The related experiments in Section 4.5 led to very promising results. We believe derivative distillation to be both a novel and high-impact contribution to future research, as it allows for an incremental definition and training of physical models, improving efficiency and the learnability of constraints by composing more than one model. Unfortunately, the reviews did not comment on this point and the scores did not seem to take derivative distillation into consideration.

From the reviews, it seems that the supervised nature of our method is a disadvantage. We do not understand why this should be the case. Although the comparison with PINN may have been misleading for some reviewers, we never claimed our method to be unsupervised. Furthermore, there are many methods in the literature, even recently published, that define and create new supervised methods to learn physical systems: NeuralODEs [1], Hamiltonian Neural Networks [2], Fourier Neural Operators [3] are just some examples of supervised methods with different architectures published in recent years, as we also report in our related works section (3). Our approach adopts the supervised learning paradigm and outperforms other supervised methods. And even PINN, in many cases.

We also would like to stress that our method is not specific to a PDE but it is rather of general application (as shown by our experiments on different PDEs). It only requires automatic differentiation to be employed. We tested it against PINNs, as they are the state-of-the-art for physical consistency, with an MLP backbone. However, DERL works also on other architectures, as we showed by using it with the Neural Conservation Laws architecture [4] in section 4.5.2.

[1] Chen, Tian Qi et al. “Neural Ordinary Differential Equations.” Neural Information Processing Systems (2018).

[2] Greydanus, Sam et al. “Hamiltonian Neural Networks.” Neural Information Processing Systems (2019).

[3] Li, Zong-Yi et al. “Fourier Neural Operator for Parametric Partial Differential Equations.” ArXiv abs/2010.08895 (2020).

[4] Richter-Powell, Jack et al. “Neural Conservation Laws: A Divergence-Free Perspective.” ArXiv abs/2210.01741 (2022).

---

### Meta-Review · Area_Chair_5uFU · 2024-12-16

**Metareview:**

The paper proposes a supervised learning approach for training neural networks by fitting their derivatives provided in a dataset. This method is evaluated on physical systems and compared with other approaches that utilize different forms of data. While the numerical results appear to favor the proposed method, all reviewers raised concerns about the fairness of these comparisons. For example, PINNs is an unsupervised approach that does not require labeled data, unlike the proposed method. This fundamental methodological difference makes direct comparisons challenging.

To improve the paper, the authors could focus on scenarios where different methods are equally applicable (e.g., OUTL and PINNs). One promising scenario mentioned is model distillation from a pre-trained model, where diverse types of data are available. However, the motivation and structure of the paper need clearer organization to enhance readability and impact. Additionally, comparisons with Sobolev learning (SOB)/gradient-augmented learning methods would provide significant value. The proposed method closely relates to SOB but uses fewer supervision signals while reportedly achieving better performance. The reason for this phenomenon is only vaguely explained. As noted by reviewers, a more systematic investigation into this phenomenon, including robustness analysis, would significantly enhance the paper’s scientific contribution.

Overall, the proposed method shows promise, but the paper in its current form is not ready for publication due to the aforementioned concerns. Addressing these points could help strengthen the work for future submissions.

**Additional Comments On Reviewer Discussion:**

During the discussion period, the authors attempted to clarify some conceptual concerns raised by reviewers. However, no substantial changes were made to the paper, such as rewriting unclear statements or providing additional numerical experiments to address the criticisms.

---

### Decision · Program_Chairs · 2025-01-22

Reject